# Decreasing extents of Archean serpentinization contributed to the rise of an oxidized atmosphere

James Andrew M. Leong [1,3 ✉], Tucker Ely[1,4,5] & Everett L. Shock[1,2]

At present, molecular hydrogen ($H_2$) produced through Fe(II) oxidation during serpentinization of ultramafic rocks represents a small fraction of the global sink for $O_2$ due to limited exposures of ultramafic rocks. In contrast, ultramafic rocks such as komatiites were much more common in the Early Earth and $H_2$ production via serpentinization was a likely factor in maintaining an $O_2$-free atmosphere throughout most of the Archean. Using thermodynamic simulations, this work quantifies the global $O_2$ consumption attributed to serpentinization during the past 3.5 billion years. Results show that $H_2$ generation is strongly dependent on rock compositions where serpentinization of more magnesian lithologies generated substantially higher amounts of $H_2$. Consumption of >2 Tmole $O_2$ yr$^{-1}$ via low-temperature serpentinization of Archean continents and seafloor is possible. This $O_2$ sink diminished greatly towards the end of the Archean as ultramafic rocks became less common and helped set the stage for the Great Oxidation Event.

[1] School of Earth and Space Exploration, Arizona State University, Tempe, AZ, USA. [2] School of Molecular Sciences, Arizona State University, Tempe, AZ, USA. [3] Present address: Lamont-Doherty Earth Observatory, Columbia University, New York, NY, USA. [4] Present address: Department of Earth and Environmental Sciences, University of Minnesota, Minneapolis, MN, USA. [5] Present address: 39 Alpha Research, Tempe, AZ, USA. ✉email: jmleong@asu.edu

The Great Oxidation Event (GOE) after the end of the Archean marks the initial oxygenation of the Earth's surface through a dramatic increase in the atmospheric $O_2$ levels from minor amounts (<$10^{-5}$ of present atmospheric levels or PAL) to much higher values ($10^{-4}$–$10^{-2}$ PAL)[1,2] around 2.4 to 2.1 billion years (Ga) ago[2–4]. While the production of $O_2$ through oxygenic photosynthesis evolved much earlier[5] and contributed to the localized accumulation of $O_2$ well before the GOE[6–11], it was only after the end of the Archean that $O_2$ began to accumulate to significant levels in the atmosphere. It is proposed that the period preceding the GOE was characterized by the dominance of geochemical and biological processes that consume $O_2$ (sinks) over its production (sources). As with an increase in $O_2$ sources, decreasing magnitudes of $O_2$ sinks in the form of reduced volatiles (e.g., $H_2$, $CH_4$) and solids (e.g., Fe(II)-bearing minerals) allow accumulation of $O_2$ in the atmosphere towards the end of the Archean. Decreases in reduced volatiles could have been brought about by the escape of $H_2$ from the atmosphere[12–15] and a transition to outgassing of more oxidized volatiles from volcanism[12,16–20]. Alternatively, recent works have tied the rise of $O_2$ to secular changes in the composition of the Earth's continental crust[21,22]. The Earth's crust has a vast potential to consume $O_2$ via the oxidation of ferrous iron in primary minerals into ferric iron in minerals that formed during rock alteration and weathering, as depicted by the reaction

$$4Fe(II)O_{(primary\ minerals)} + O_2 \rightarrow 2Fe(III)_2O_{3(secondary\ minerals)}. \quad (1)$$

The transition from a continental crust composed dominantly of Fe-rich mafic rocks to one that is composed mostly of Fe-poor felsic rocks during the Archean resulted in a decrease in the reductive efficiency of the Earth's crust that could have facilitated the initial oxygenation of Earth's surface[21].

The absence of $O_2$ does not impede the oxidation of rocks. At great extents of water-rock interactions such as those occurring in deeper aquifers where $O_2$ and other dissolved oxidants are exhausted, ferrous iron in the protolith is oxidized to ferric iron in the alteration assemblages and, in turn, water is reduced into $H_2$ as depicted by the reaction

$$Fe(II)O_{(primary\ minerals)} + H_2O \rightarrow Fe(III)_2O_{3(secondary\ minerals)} + H_2, \quad (2)$$

supplying an additional sink for $O_2$ through

$$O_2 + 2H_2 \rightarrow 2H_2O. \quad (3)$$

The amount of $H_2$ that can be generated through reaction (2) is far less dependent on the abundance of Fe present in the rock than the bulk composition[23]. An extreme example is the hydrous alteration of ultramafic rocks, which is known as serpentinization because most of the original minerals are replaced by serpentine. Serpentinization produces some of the most $H_2$-rich fluids on Earth that contribute to much of the modern geological supply of abiotic $H_2$[24]. Alteration of mafic rocks such as basalts also yields $H_2$ but of lower quantity than those generated in ultramafic-hosted environments[25,26]. Recent experimental work shows that the alteration of felsic rocks at hydrothermal conditions can also yield a significant amount of $H_2$[27]. However, it is unknown if this extends to the low-temperature conditions present in most continental aquifers. Overall, locations, where substantial amounts of abiotic $H_2$ are actively generated, are limited in modern Earth settings to where ultramafic rocks can interact with water such as in slow-spreading ridges, passive margins, subduction zones, and in uplifted ultramafic bodies on continents[24]. In contrast to the modern Earth, ultramafic rocks such as komatiites were much more widespread early in Earth's history due to higher mantle temperatures[28], implying that $H_2$ production through serpentinization would have been more prevalent.

Several authors propose that the decreasing flux of serpentinization-generated volatiles, via the progressive loss of ultramafic rocks exposed at the surface as the mantle cooled through the Archean, could have helped facilitate the GOE[22,29–32]. However, it is unclear if this trend is sufficient to permit significant accumulation of $O_2$ in the atmosphere as the magnitude of the decrease in the flux of serpentinization-generated volatiles has never been quantified for the period leading to the GOE. The purpose of this communication is to report the results of such computations. This work combines thermodynamic simulations of water-rock interaction with mass-transport calculations to estimate the flux of $H_2$ generated through the low-temperature alteration of Fe-bearing igneous rocks. Alteration simulations were conducted on 9,414 rocks of variable compositions taken from the GEOROC database[33] (Source Data S1, http://georoc.mpch-mainz.gwdg.de/georoc/). Rock compositions include picrites, komatiites, and other ultramafic rocks that were likely common during the Archean. Calculations are focused on low-temperature conditions (25 °C) to simulate the ambient conditions prevalent in most of Earth's aquifers. Consequently, past global $O_2$-consumption rates attributed to low-temperature serpentinization can be estimated, allowing quantitative assessment of whether the decreasing extent of komatiite volcanism towards the end of the Archean set the stage for the Great Oxidation Event.

## Results and Discussions

**Serpentinization is the key to significant $H_2$ production.** Although $H_2$ production is tied to the oxidation of ferrous iron in rocks (reaction 2), the hydrous alteration of igneous rocks with similar ferrous iron content does not always yield similar amounts of $H_2$. As an example, basaltic and ultramafic rocks have similar ferrous iron contents (6–14 weight percent, wt%, FeO) and, seemingly, the hydrous alteration of both types of rocks should generate similar amounts of $H_2$ via reaction (2). However, fluids that are most enriched in $H_2$, such as those venting at the Lost City and Rainbow hydrothermal fields[34–36], are commonly associated with ultramafic-hosted environments with some inputs from intermingling mafic rocks[35]. Ultramafic-hosted hydrothermal fluids can attain >10 $m$molal $H_2$ that is several times more concentrated than most basalt-hosted hydrothermal fluids[37]. This contrast is particularly distinct in lower temperature environments such as those occurring in continents. Ultramafic-hosted hyperalkaline groundwater can attain >1 $m$molal $H_2$ (e.g., Oman ophiolite[38]) while most basalt-hosted fluids rarely exceed 0.01 $m$molal $H_2$ (e.g., Columbia river basalt[39]). The greater potential for ultramafic rocks to generate $H_2$ during alteration is attributed to their bulk compositions[23]. Previous thermodynamic simulations have related the compositions of a limited number of reacting rocks, mostly peridotites, and basalts, with the redox potentials of resulting hydrothermal fluid[26,40,41]. Using the expansive GEOROC database, thousands of simulations conducted in this work provide an inclusive assessment of the $H_2$-generation potentials of rocks with compositions ranging from ultramafic to mafic. Equilibrium simulations yield aqueous compositions that are consistent with those measured from end-member low-temperature hyperalkaline fluids sampled from ultramafic bodies in ophiolites[42], that are modern analogs of Archean ultramafic-hosted fluids. The approach to equilibrium even under ambient conditions is possible as the timescales required to attain reduced and hyperalkaline compositions may involve several thousands of years[43].

Results of simulations depicted in Fig. 1a, b, and Source Data S2 show the calculated amounts of $H_2$ generated during

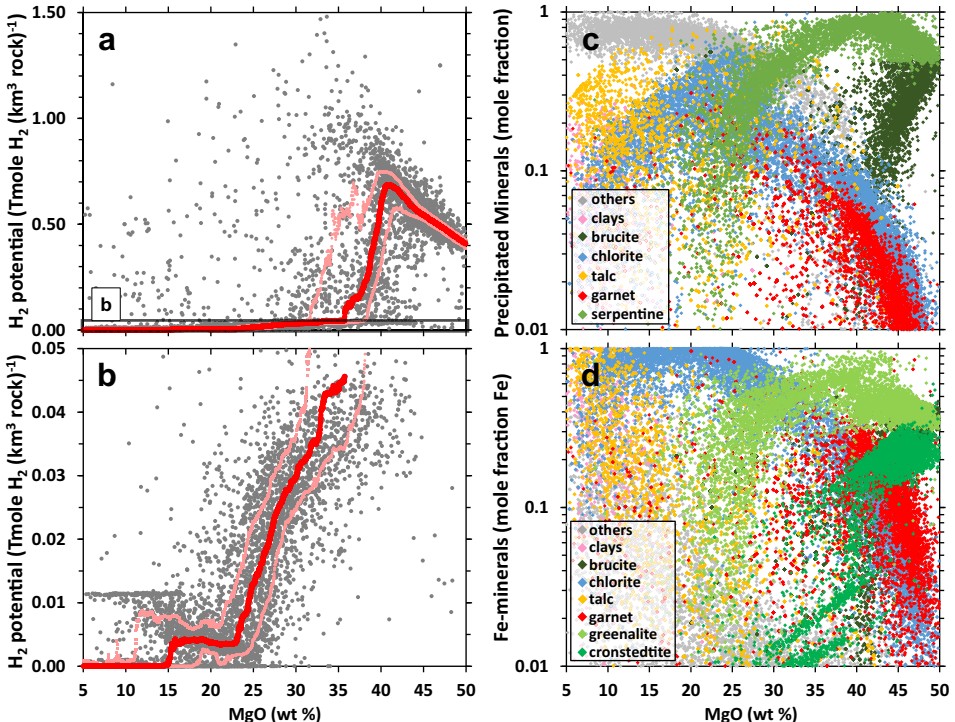

**Fig. 1 Compositional controls on H₂ generation during low-temperature alteration. a** The H₂-generation potential (Tmole H₂ (km³ rock)⁻¹) during hydrous alteration of 9,414 Fe-bearing igneous rocks with compositions ranging from ultramafic (right, high MgO%) to basaltic (left, low MgO%). H₂-generation values are from simulations depicting low-temperature (25 °C) conditions at a water-to-rock ratio = 1. **b** Close-up of the H₂-generation potentials of rocks within the black box in (**a**). Red and upper and lower pink symbols in (**a**) and (**b**) depict rolling median, 95th, and 5th percentiles, respectively. Rolling values for a rock sample of given composition (e.g., 36.5 MgO wt%) are calculated from the H₂-generation potentials of all rock samples that have MgO contents within 1 wt% of the given value (e.g., 35.5–37.5 MgO wt%). **c** Minerals (in mole fraction) calculated to form in the simulations. **d** Distribution of Fe among the secondary phases.

low-temperature (25 °C) alteration of Fe-bearing igneous rock types at a water-to-rock ratio of 1 (i.e., alteration of 1 kg of rock in 1 kg of water). Predictions are summarized for hydrous alteration of 9,414 Fe-bearing igneous rocks from the GEOROC database with compositions ranging from those that are ultramafic, i.e., rocks that have high MgO content such as peridotites and komatiites at the right side of Fig. 1a, b, to those relatively depleted in MgO (left side of Fig. 1a, b) such as picrites and basalts. Further details on the relationship between the composition of reacting rocks and their H₂-generation potentials are shown in Fig. S1. All of these rocks have similar FeO contents (6–14 wt%, Fig. S1a). Basalts and picrites originate from smaller extents of mantle melting while komatiites were generated by larger extents of melting that occurs at higher temperatures, which usually resulted in rocks with higher Mg and lower Si (Fig. S1b), Ca (Fig. S1c), and Al (Fig. S1d) contents[44]. Most of the peridotites exposed in the seafloor and in the continents are uplifted harzburgite, which are the residual rocks that remained after melting of the mantle and hence are characterized by high Mg content and low Si and Al values. As shown in Fig. 1a, there is a marked decrease in the H₂-generation potentials between rocks with MgO content lower and higher than 35 wt%, even among rocks without much difference in their FeO abundances. Model results also depict a gradual decrease in H₂-generation potentials as the reacting compositions become less Mg-rich (towards 20 wt% MgO), as shown in Fig. 1b. These results corroborate natural observations that ultramafic-hosted fluids are more enriched in H₂ than those hosted in basalts. The generation of highly reduced fluids during alteration of Mg-rich rocks is also

supported by the presence of accessory minerals only stable in reduced conditions such as NiFe alloys (awaruite) found in some altered ultramafic rocks[45].

The key to the transitions in the H₂-generation potentials during rock alteration documented in Fig. 1a, b is the formation of serpentine. Mg-rich rocks favor the formation of serpentine during rock alteration, as shown in Fig. 1c, which depicts the overall abundances of secondary minerals (in mole fraction) calculated to form. In the models, initial stages of serpentinization, characterized by high water-to-rock ratios, typically leads to the formation of Fe-bearing serpentine minerals that host ferric iron: hisingerite [(FeIII)₂Si₂O₅(OH)₄] and cronstedtite [(Fe(II)₂ (FeIII))(Fe(III)Si)O₅(OH)₄]. In contrast, serpentinization at rock-dominated conditions favors the formation of greenalite [(Fe(II)₃Si₂O₅(OH)₄], which is a Fe-bearing endmember component of serpentine that only hosts ferrous iron. Simulations incorporate an ideal-site solid solution model that involves the above Fe end-members and Mg end-member serpentine (chrysotile, Mg₃Si₂O₅(OH)₄). Model results are consistent with analysis of natural serpentinites where higher Fe(III)/ΣFe values were measured in serpentine thought to form at higher water-to-rock ratios than those that precipitated at rock-dominated conditions[46]. As shown in Fig. 1d, which depicts the distribution of Fe in the secondary assemblages predicted to form at a low water-to-rock ratio of 1, the Fe end-member serpentine favored to form is greenalite followed by cronstedtite. The amount of hisingerite that formed at the low water-to-rock conditions simulated in this work is insignificant and plots below the range shown in Fig. 1d. Aside from the water-to-rock ratio, the

composition of reacting rocks also controls the distribution of Fe in the precipitating serpentine. Cronstedtite is most favored to form during alteration of rocks with high MgO content (> 40 wt%, see Fig. 1d). As the MgO content of the reacting rock decreases from 40 to 20 wt%, the formation of cronstedtite is drastically reduced. In contrast, the formation of greenalite generally remains constant and only decreases (along with serpentine) when the MgO content of the reacting rocks is between 20 and 25 wt%. The decreasing potential to form cronstedtite relative to greenalite results from the decreasing MgO and increasing $SiO_2$ contents of the rocks as cronstedtite has a lower Si content (1 mole per formula unit) than greenalite (2 moles per formula unit)[47]. The decrease in the cronstedtite content in the precipitating serpentine with decreasing MgO means that less Fe will be oxidized resulting in less $H_2$ production, as shown in Fig. 1b, d.

Another Fe(III)-bearing phase, andradite garnet [$Ca_3(FeIII)_2$-$Si_3O_{12}$], is also favored to form during alteration of Mg-rich rocks and provides additional pathways for $H_2$ formation[48]. Not all Mg-rich rocks favor the formation of garnet, which also depends on their CaO content, resulting in highly variable $H_2$-generation potentials for rocks with MgO contents between 30 and 40 wt%. In contrast, rocks that are poorer in Mg (MgO <20%) do not favor serpentine and garnet formation during rock alteration. Instead, alteration of these Mg-poor but more Si- and Al-rich rocks stabilizes minerals such as chlorite, talc, and clay minerals (Fig. 1c, d). Most of the iron mobilized from the primary minerals is incorporated into these secondary minerals without much oxidation because these minerals preferentially accommodate ferrous iron into their crystal structures. Consequently, less $H_2$ is generated as the iron oxidation process depicted by reaction (2) occurs at lesser extents compared to conditions where serpentine and garnet form extensively. Overall, results show that non-redox-sensitive components of rocks (e.g., Mg, Si, Al, Ca) determine the distribution and fate of Fe during secondary mineralization, and therefore the redox processes that generate reduced volatiles during fluid-rock interactions. Further discussions on how various types of mineralization control $H_2$ production can be found in the supplementary document and are illustrated in Fig. S1.

A statistical summary of the $H_2$-generation potentials of rocks with MgO content <10, >45, and those ranging from 10–45 wt%, at intervals of 5 wt%, subjected to various water-to-rock ratios (100, 10, 1, 0.2) is shown in Fig. 2. Full data can be found in Source Data S2. The distribution of $H_2$-generation potentials of rocks within a compositional group is further shown in Fig. S2. Rocks with MgO content >35 wt% have the potential to generate the most $H_2$ but of highly varying amounts owing to their differing capacities to stabilize various Fe(III)-bearing phases during hydrous alteration. Rocks with similar MgO content can have variable $SiO_2$ contents, which lead to varying potentials to precipitate Fe-bearing secondary phases (see Fig. S1b). Those with higher $SiO_2$ contents favor formation of greenalite relative to cronstedtite. Hence, less $H_2$ can be generated as Fe(II) from primary minerals is mobilized into greenalite unoxidized[47,49]. Rocks with MgO content between 20–35 wt% have potentials to generate moderate amounts of $H_2$ only at higher water-to-rock ratios. Rocks with MgO content <20 wt% do not yield high $H_2$ at any water-to-rock ratios simulated in this work.

**Decreasing $O_2$ sink via continental serpentinization during the past 3.5 Ga.** In the modern Earth, ultramafic rocks comprise only ~0.2% of the continental surface[50], mostly as uplifted harzburgites and other peridotitic rocks in ophiolites and orogenic massifs. Volcanism of ultramafic lavas is rare throughout most of the Proterozoic but is known to have been more prevalent during the Archean[28,51]. In the Archean, ultramafic rocks such as komatiites are estimated to have comprised a significant component of the continental crust (~10–20% during the early Archean[52,53]). This elicits scenarios of substantial fluxes of serpentinization-derived $H_2$ that can maintain an atmosphere with elevated $H_2$ levels during the Archean[54].

The global $O_2$-consumption potentials via $H_2$ production during low-temperature serpentinization during the past 3.5 Ga were calculated using Eqs. (4) and (5) (see methods) following estimates of $H_2$ generation ($x_{Fe}$, Tmole $H_2$ $km^{-3}$) shown in Fig. 1 and summarized in Fig. 2. In these calculations, we assume a serpentinization rate ($r_{Fe}$) of $10^{-6}$ km $yr^{-1}$, a maximum value for low-temperature ophiolitic aquifers with low reactive surface area[55]. For the extents of ultramafic rocks in continents ($a_{Fe}$, $km^2$), we used the average distribution and continental presence calculated by Greber et al.[53] and Dhuime et al.[56], respectively. Tang et al.[52] disagree with Greber et al.[53] on the bulk composition and timing of the transition from mafic to felsic crust during the Archean. However, both studies are consistent in their estimates of past ultramafic distribution, and both estimate the significant presence of komatiites in continents during the early Archean (10–20%) and a greatly diminished distribution towards the end of the Archean. Multiplying all the above values together yields the annual global $H_2$ generation (Tmole $H_2$ $yr^{-1}$) and consequently the annual global $O_2$ consumption (Tmole $O_2$ $yr^{-1}$). Results of calculations are depicted in Fig. 3, and Source Data S3, which shows the $O_2$-consumption potentials of $H_2$ produced during alteration of rocks with given ranges of MgO content. We only show results for these relatively Mg-rich rocks as the continental compositional estimates by Greber et al.[53] pertain to komatiites and perhaps other ultramafic bodies, which we used as a proxy for Mg-rich rocks. Analyses of remnant komatiites in cratonic belts yield MgO values ranging from ~18 wt% in evolved spinifex-textured lavas to ~45 wt% in olivine-rich cumulates[51]. Binning the wide range in the MgO content of rocks used in our calculations to every 5 wt% accounts for uncertainties in the compositions of komatiites when they are first exposed during the Archean. While Greber et al.[53] and Tang et al.[52] also estimated the amount of basalts present in continents during the past 3.5 Ga, results of alteration simulations for rocks with MgO content <20 wt%, encompassing all basalts and most picrites, yield negligible potentials to generate $H_2$ and consume $O_2$ (Figs. 1 and 2), and thus are not included.

As shown in Fig. 3, the global $O_2$ consumption arising from the alteration of various Fe-bearing rocks with MgO content >20% greatly decreased at ~2.5 Ga ago, and this decrease is directly attributed to the decrease in the extent of exposures of Mg-rich rocks in the continents. A marked decrease in komatiite distribution from ~7% to ~1% of continents from 3.0 to 2.5 Ga ago[53] caused a sevenfold decrease in the $O_2$-consumption potentials during this time interval and contributed to the rise of $O_2$ by the end of the Archean. This monotonic decrease may not be characteristic of the late Archean if the occurrences of several 2.7 Ga greenstone belts[28,51] are a consequence of increased komatiite volcanism at this time period and not solely a result of exceptional preservation. Nevertheless, komatiite occurrence is significantly diminished by the end of the Archean and thus would not change model outcomes depicting the Archean-Proterozoic transition shown in Fig. 3. Note that results shown in Fig. 3 are solely dependent on the ultramafic distribution in continents ($a_{Fe}$ component of Eq. 4) as the rate ($r_{Fe}$), and $H_2$-generation potential ($x_{Fe}$) are assumed to be constant across time for each investigated Fe-bearing igneous rock. The alteration rate, $H_2$-generation potentials, and therefore the $O_2$-consumption potentials could have larger values for the

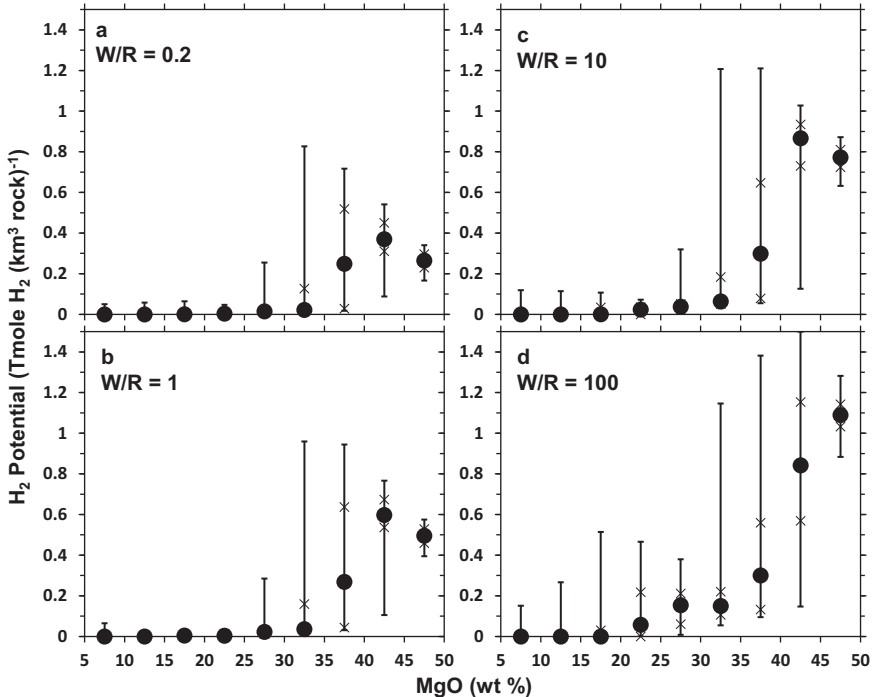

**Fig. 2 Summary of H$_2$-generation potentials of various rock compositional groups.** Median (circle), 25th–75th percentile (lower and upper Xs, respectively), and 5th–95th percentile (lower and upper ticks, respectively) of H$_2$-generation potentials yielded by simulations of alteration at various water-to-rock ratios: **a** 0.2, **b** 1, **c** 10, **d** 100 for various 5 wt % ranges of MgO content (e.g., 30–35 wt%).

Archean due to the warmer conditions and more acidic or reactive starting fluids derived from an atmosphere with higher $p$CO$_2$[57]. Thus, results of our calculations should be considered conservative and decreases in the O$_2$-consumption values during the past 3.5 Ga could be even more dramatic than those depicted in Fig. 3. Simulation results also show that alteration of rocks with MgO content >35 wt% has the greatest potential to consume O$_2$ (Fig. 3a, b). Median potentials for rocks with MgO content ranging from 20 to 35 wt% are far lower (Fig. 3d–f). Those with MgO content between 30 and 35 wt% mostly have similar potentials compared with those with less Mg but a few rocks within this compositional range have similar potentials to consume O$_2$ as those with higher MgO content (see upper thinner lines in Fig. 3c).

From 1.0 Ga ago to the present, ultramafic rocks are estimated by Greber et al.[53] to comprise at most 0.2% of the continents, consistent with abundances determined by analysis of global present-day geological maps[50]. Median values yielded by models show global H$_2$ production ranging from 0.08 to 0.18 Tmole yr$^{-1}$ and correspondingly, O$_2$ consumption ranging from 0.04 to 0.09 Tmole yr$^{-1}$ for the past 1.0 Ga (Fig. 3 and Source Data S3). These values are consistent with other estimates of modern-day outgassing rates in continental serpentinizing environments (0.02–0.18 Tmole H$_2$ yr$^{-1}$)[58]. Model results are also within range, but on the lower bound, of estimates of global H$_2$ production associated with serpentinization of oceanic ultramafic rocks (0.17–0.70 Tmole H$_2$ yr$^{-1}$)[25,31,59–61]. These values are insignificant relative to the overall modern global O$_2$ sources and sinks (>15 Tmole O$_2$ yr$^{-1}$ (refs. [16,31,32])). The magnitude of net O$_2$ production on the early Earth remains unconstrained though recent analysis of the carbon isotopic record and extent of organic carbon burial show that Archean primary productivity may have been comparable to modern values[62]. If Archean net O$_2$ production was similar to that of the present (>15 Tmole O$_2$ yr$^{-1}$), global O$_2$ consumption via serpentinization of Archean continents alone would have been a major sink for oxygen given

that consumption of >2 Tmole O$_2$ yr$^{-1}$ (~13% of sources) and, in some extreme cases, >4 Tmole O$_2$ yr$^{-1}$ (~27%) is possible between 3.5 and 3.0 Ga ago if the reacting rocks are Mg-rich (MgO >35 wt%, Fig. 3). Consequences of variable continental extents as well as additional contributions from seafloor serpentinization are discussed below.

**Effect of variable extent of ultramafic exposure in continents.** The calculations summarized above use estimates of past continental mass from Dhuime et al.[56], which predicts that the amounts of continents throughout the Archean are intermediate to more extreme values estimated by earlier studies[63–66]. To further explore the consequences of these variations, we calculated the potential for consuming different levels of O$_2$ via serpentinization given variable extents in the exposures of Mg-rich rocks in continents. Outcomes of calculations are shown in Fig. 4a which depicts the proportion of simulations that results in the consumption of >2 Tmole O$_2$ yr$^{-1}$ given the abundance of Mg-rich rocks (in km$^2$) and an alteration rate of 10$^{-6}$ km yr$^{-1}$.

Given the range in the exposure of Mg-rich rocks present at the mid-Archean (3.0 Ga, light grey region in Fig. 4a), up to 100, 90, and 46% of models simulating rocks with MgO >45, 40–45, and 35–40 wt%, respectively, result in the consumption of >2 Tmole O$_2$ yr$^{-1}$. Only up to 17% of the models involving rocks with MgO content between 30–35 wt% result in a similar amount of O$_2$ consumption. Consumption of >2 Tmole O$_2$ yr$^{-1}$ is still possible if most Archean komatiites have MgO content <30 wt% but are less likely as only ~5% of simulations result in H$_2$ production that can offset this rate of net O$_2$ production. The above values depict results of models assuming maximum exposure using the continental growth model of Armstrong[63] who argues for present-day extent of continents throughout most of the Archean. Using the recent model of Dhuime et al.[56] yields lower but still significant values. In contrast, smaller masses of continents proposed by other works[64–66] would yield lower likelihoods of H$_2$ generation from continental serpentinization during the Archean.

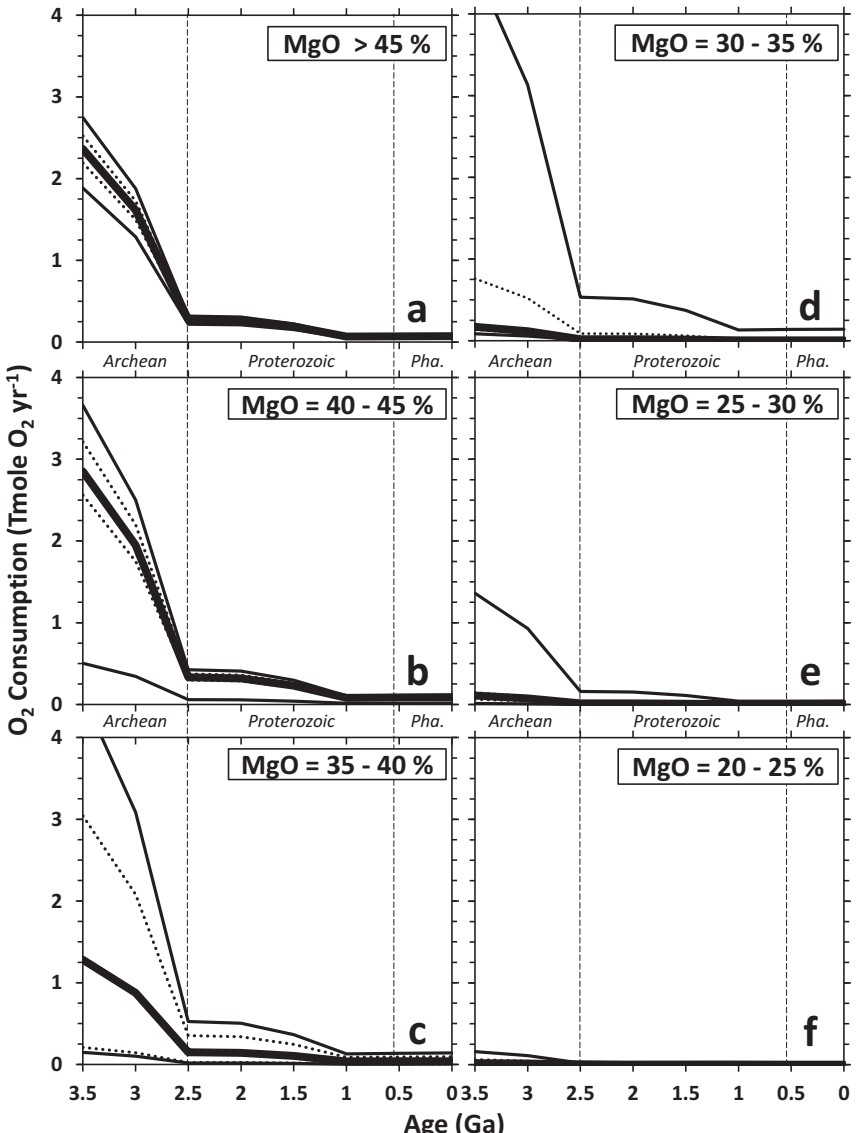

**Fig. 3 O₂ consumption via continental serpentinization during the past 3.5 Ga.** Global consumption rate for $O_2$ (Tmole yr⁻¹) was calculated assuming an alteration rate of $10^{-6}$ km yr⁻¹ for rocks belonging to various compositional groups: **a** MgO >45, **b** 40–45, **c** 35–40, **d** 30–35, **e** 25–30, **f** 20–25 wt%. Extents of continental mass and exposure of komatiites are from Dhuime et al.[56] and Greber et al.[53], respectively. Bold curve depicts median values. The lower and upper dotted curves represent 25th and 75th percentiles of model results, respectively, while the lower and upper thinner curves depict 5th and 95th percentiles of model results, respectively.

These results imply that for continental serpentinization to be a significant sink for $O_2$ throughout most of the Archean, continental presence must be significant (i.e., at least 60% of the present-day value at 3.0 Ga[56]) and occurrence of Mg-rich (MgO >35 wt%) rocks must be extensive. Thus, the majority of $H_2$ production occurs in Mg-rich ultramafic bodies such as peridotite massifs or komatiite flows with significant olivine cumulates[46]. Cumulates are often associated with deeper portions of komatiite flows but fluids are known to infiltrate and interact with deep-seated rocks[58]. Furthermore, deep-seated rocks can readily react with groundwater during and after continental emplacement like in modern ophiolitic bodies where peridotites are exposed to surface conditions and enable the generation of most present-day $H_2$-rich, hyperalkaline springs. Results of additional calculations assuming slower rates ($10^{-6.5}$ km yr⁻¹) are depicted in Fig. S3, which shows that a much smaller proportion of simulations result in significant $H_2$ production and

$O_2$ consumption. Therefore, for serpentinization to be a significant source of reduced gas throughout most of the Archean, the rate of serpentinization should be at least $10^{-6}$ km yr⁻¹. While this represents an upper limit on rates perceived to occur in modern low-temperature ultramafic aquifers[55], such a rate could have been normal under warmer and more dynamic Archean surface conditions.

Simulations by Kadoya et al.[20] show that in addition to volcanic volatiles, additional flux of reduced gas generated via serpentinization amounting to consumption of at most 2 Tmole $O_2$ yr⁻¹ would result in oxidation of the atmosphere by 2.4 Ga ago. Larger fluxes from serpentinization such as 3 Tmole $O_2$ yr⁻¹ could have delayed the GOE to 1.9 Ga ago[20]. Assuming the amount of Mg-rich rocks present in continents ~3.0 Ga ago, up to 35% or 80% of models result in consumption of >3 Tmole $O_2$ yr⁻¹ via alteration of rocks that have MgO content 35–40 wt% or >40 wt%, respectively (Fig. 4b). A significantly lower number

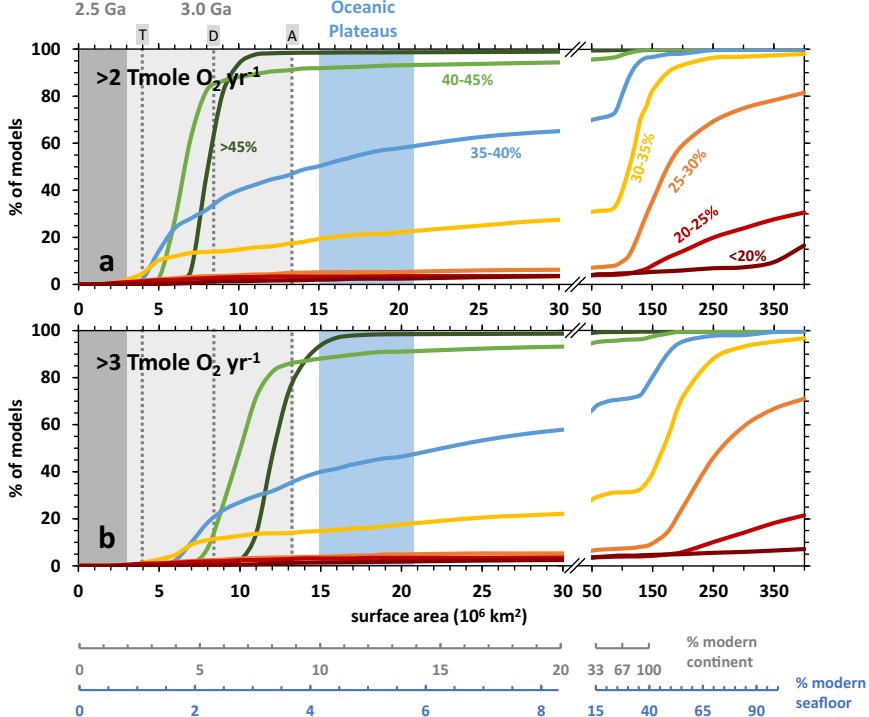

**Fig. 4 O$_2$ consumption vs extents of ultramafic rocks.** Potentials for consuming more than 2 (**a**) or 3 (**b**) Tmole O$_2$ yr$^{-1}$ as functions of the extent of Fe-bearing rocks exposed in either continents or seafloor (in km$^2$). Scales at the bottom compare given values to present-day continents and seafloor extents. Results are from models assuming serpentinization rate = 10$^{-6}$ km yr$^{-1}$. Light and dark grey shaded areas depict estimated ranges in the extent of ultramafic rocks exposed in continents 3.0 and 2.5 Ga ago, respectively. Vertical dashed grey lines depict maximum extent of ultramafic or komatiite exposures at 3.0 Ga based on compositional estimates of Greber et al.[53] and continental growth models of (A) Armstrong[63], (D) Dhuime et al.[56], and (T) Taylor and McLennan[65]. Other continental growth models[64,66] yields much lower maximum ultramafic extents. The blue shaded area represents the extent of oceanic plateaus at the present-day (~4–6% of the seafloor)[71]. Curves of various colors depict the results of models simulating rocks belonging to different compositional groups as indicated by the MgO contents.

of models results in scenarios that can offset the same amount of O$_2$ during alteration of rocks with lower MgO contents. Thus, if the estimated amount of Mg-rich ultramafic rocks (MgO >35 wt %) present in continents at ~3.0 Ga remained by ~2.4 Ga ago, then the abundance of reduced volatiles released from both serpentinization and volcanic outgassing would have delayed the GOE to a much later period. However, the extent to which ultramafic rocks were present in continents declined significantly from 3.0 to 2.5 Ga ago[28,52,53], as shown by the dark grey shaded regions in both plots in Fig. 4 where the amount of ultramafic rocks present in continents ~2.5 Ga ago should be not more than 3 million km$^2$. Consequently, the decreased presence of ultramafic rocks in continents would have resulted in lower potentials for an O$_2$ sink. As shown in Fig. 4, close to 0% of models encompassing all compositional ranges explored by this work result in consumption of >2 Tmole O$_2$ yr$^{-1}$ by 2.5 Ga ago, ensuring the accumulation of O$_2$ in the atmosphere and setting the stage for the Great Oxidation Event.

**Contributions from seafloor serpentinization.** In addition to H$_2$ produced by serpentinization in continents, significant amounts of H$_2$ can be generated through water-rock interactions in the subseafloor. Modern seafloor serpentinization contributes as much, perhaps more, H$_2$ than that generated via continental serpentinization[58]. While the extent of ultramafic exposures in the seafloor during the Archean is poorly known, the importance of additional H$_2$ generated from seafloor processes warrants preliminary discussion.

A hotter mantle during the Archean would generate Mg-rich oceanic crust that could have favored H$_2$ production. However, a hotter ambient mantle that yields erupting lavas with MgO contents between 15 and 25%, higher than present seafloor basalts, would still not generate enough H$_2$ to result in significant O$_2$ drawdown (Fig. 4a). Rocks with MgO contents of 25–30%, which require a much hotter mantle source[44], would still need to comprise >150 million km$^2$ of the seafloor (~40% of present-day) for the likelihood of offsetting >2 Tmole O$_2$ yr$^{-1}$ to exceed 50% (Fig. 4a). Variable thickness of highly impermeable sedimentary layers can cover and prevent extensive exposures of the igneous oceanic crust, especially that which is older, and control communication between seawater and basement rocks[67]. Less pervasive sedimentation in the Archean seafloor would likely favor greater extents of water-rock interactions and higher potentials to produce H$_2$.

Most of the outgassing of H$_2$ in modern seafloor environments is limited to slow-spreading environments where tectonically exposed residual peridotites readily interact with seawater[60]. Faster spreading rates[68] and thicker oceanic crust[69] that likely characterized the Archean would not favor emplacement of deep-seated Mg-rich residual peridotites and cumulate rocks into the surface. Instead of spreading centers, Mg-rich rocks are believed to have erupted during the Archean in hotter plume-generated settings such as oceanic plateaus formed by large igneous provinces[70]. Oceanic plateaus can be extensive and are estimated to comprise ~5% of present-day seafloor (~18 * 10$^6$ km$^2$)[71]. If similar extents (15–21 * 10$^6$ km$^2$) of oceanic plateaus comprised of rocks with MgO contents <30% were present during the Archean, the likelihood of generating H$_2$ that can offset >2 Tmole O$_2$ yr$^{-1}$ is still low (<5% of models, blue field in Fig. 4). A similar extent of exposure of rocks with MgO content >35% would have

resulted in much higher potentials to offset a similar rate of net $O_2$ production. However, Mg-rich cumulate komatiites typically comprise a minor component of eruptive komatiite flows that are likely to have been compositionally heterogeneous. $H_2$-rich seeps could have been sourced from deep-seated aquifers in komatiitic bodies where interactions with cumulate rocks are more likely. Results of calculations of $H_2$ outgassing in a heterogenous igneous province comprised of variable mixtures of mafic (MgO <10%) and relatively more Mg-rich lithologies are shown as colored symbols and curves in Fig. S2a,b, respectively. As $H_2$ production in rocks with MgO >35% is far more substantial than in rocks with lower Mg contents, at least 40% (dark blue-violet curves in Fig. S2b) of a given quantity of fluids present in mafic-ultramafic igneous provinces encompassing 18 km$^2$ of the seafloor (orange line in Fig. S2b) would need to interact with rocks with MgO >35% to consume >2 Tmole $O_2$ yr$^{-1}$. A hotter mantle on the early Earth would likely result in more extensive occurrences of plume-generated igneous bodies than present (i.e., >18 km$^2$), which would increase the likelihood of offsetting production of >2 Tmole $O_2$ yr$^{-1}$.

**Cessation of Archean serpentinization as a driver for the GOE.** This work quantifies the extents of ultramafic presence in continents and the seafloor required for significant outgassing of $H_2$ during low-temperature serpentinization that would have helped maintain an $O_2$-free atmosphere throughout most of the Archean. Recent constraints on the composition[52,53] and extent[56] of Early Earth continents yield scenarios where outgassing of serpentinization-derived $H_2$ that can offset $O_2$ production of >2 Tmole yr$^{-1}$ is possible. Similar or greater extents of $H_2$ production via serpentinization of the Archean seafloor are possible but remain poorly constrained and may depend on the extents of plume-generated oceanic plateaus composed of Mg-rich ultramafic rocks. Those with >35% MgO are more likely to generate significant fluxes of $H_2$ that can result in larger sinks for $O_2$. Whereas they are abundant throughout most of the early- and mid-Archean, exposures of ultramafic rocks diminish significantly by the end of the eon. Almost all our simulations yield close-to-zero potentials to consume significant amounts of $O_2$ via $H_2$ production from rock alteration by the end of the Archean and help set the stage for the GOE.

Additional constraints on the extents of ultramafic presence in continents and the seafloor will refine global $H_2$-generation and $O_2$-consumption rates on the Early Earth. Quantifying $H_2$ outgassing during both low- and high-temperature serpentinization of ultramafic rocks present in plume-generated oceanic plateaus, as well as other marine settings like ocean ridges, passive margins, and subduction zones, during the Archean will yield more comprehensive global models of serpentinization-derived $H_2$ outgassing. Serpentinization is only one of the many sources of reductants to the Earth's surface (e.g., volatiles that are volcanically and microbially derived). We hope that our simulations will ultimately contribute to an integrated model that incorporates evolution of various redox sources and sinks, many of which can be facilitated by the secular change in the composition of Earth's continents and seafloor, that will yield forward models predictive of surface redox conditions.

## Methods
**Overview**. The annual global outgassing of $H_2$ (mole $H_2$ yr$^{-1}$) derived from the serpentinization of komatiites and other Fe-bearing igneous rocks at a given point of Earth's history, $H_{2,Fe}$, can be described by the equation

$$H_{2,Fe} = a_{Fe} r_{Fe} x_{Fe} \qquad (4)$$

where

$a_{Fe}$ = extent of Fe-bearing igneous rocks distributed in the continents or seafloor (km$^2$),

$r_{Fe}$ = rate of rock alteration (km yr$^{-1}$), and

$x_{Fe}$ = the $H_2$-generation potential of a given volume of Fe-bearing igneous rocks (mole $H_2$ km$^{-3}$).

As shown by reaction (3), consuming a mole of $O_2$ would require two moles of degassed $H_2$, which means that the global $O_2$ consumption can be estimated via

$$O_{2,Fe} = 0.5 H_{2,Fe}. \qquad (5)$$

**Calculation of the $H_2$-generation potential of a given volume of rock ($x_{Fe}$).** The amount of $H_2$ that can be generated through the serpentinization of a given volume of rock, given by $x_{Fe}$ in Eq. (4), can be variable and will depend on the compositions of the reacting rock and fluid, as well as the extent of the rock alteration process. Simulations of hydrous alteration of 9,414 Fe-bearing igneous rocks of variable compositions were conducted using the reaction-path code EQ3/6[72] together with a customized thermodynamic database (see below) to calculate $x_{Fe}$ values. Automation of rock alteration simulations and data processing was conducted following ref. [73]. These calculations simulate the hydrous alteration of a rock by a fluid and determine the compositions of coexisting solid phases and fluid constituents attained at thermodynamic equilibrium at each step of overall progress in rock alteration. Our models account for $H_2$ generated at various extents of water-rock reaction, quantified as the water-to-rock ratio. An increase in reaction progress is analogous to a decrease in the water-to-rock ratio, as the reacting water encounters more rock while infiltrating deeper into the subsurface. Active serpentinization-generated $H_2$-rich seeps are known to be products of rock-dominated systems (i.e., low water-to-rock ratios[35,42,74]). In addition, we found that at water-to-rock ratios <100 the simulation results show that the compositions of fluid, gas, and solid phases generated during the alteration of a given rock sample would be similar despite reaction with fluids of variable starting dissolved $O_2$ concentration (see Fig. S4). At low water-to-rock ratios, alteration products will be predominantly dictated by the composition of the reacting rock rather than the reacting fluid. Given these constraints on serpentinization-generated fluids, we used the results of simulations at low water-to-rock ratios (<100) to estimate past fluxes for $H_2$. At water-to-rock ratio = 0.2, the reacting fluid is mostly consumed to form hydrous minerals and the simulations were terminated by this point in the overall rock-alteration progress.

We did not include dissolved $CO_2$ in our idealized fluids owing to scarcity of information on mineral carbonation reactions in the deep past. However, note that $pCO_2$ could have been much higher in the Archean[57]. Consequently, the pH of meteoric-derived fluids infiltrating the continental subsurface could have been lower[75] and affected reaction rates. Moreover, while the inclusion of $CO_2$ in the reacting fluid can permit predictions of the amount of $CH_4$ generated, which is another sink for $O_2$, generation of abiotic $CH_4$ during low-temperature serpentinization has been questioned[76,77]. Future work constraining the concentrations of dissolved $CO_2$, as well as other solutes in the reacting fluid, would refine model results. Aside from dissolved $CO_2$, the starting dissolved sulfate[73] and Si[47] concentrations of reacting fluids can also influence the overall process of rock alteration. It has been argued that an elevated Si concentration can suppress $H_2$ production[47]. However, the extent to which $H_2$ production is suppressed at variable water-to-rock ratios (i.e., water-dominated vs rock-dominated conditions) is unknown. It is likely that at low water-rock conditions (e.g., water-to-rock ratio = 1), which is the focus of this work, the compositions of the reacting rock exert more influence than the starting compositions of the reacting fluids. This is exemplified by results of calculations depicted in Fig. S4 which shows similar $H_2$ potential for a given rock composition at low water-to-rock conditions despite interaction with fluids that have variable starting dissolved $O_2$ content. Future simulations can disentangle the relationship between $H_2$ generation, water-to-rock ratios, and the starting composition of reacting groundwater or seawater.

These idealized calculations allow us to adequately track changes in fluid chemistry during rock alteration as demonstrated by several studies on various sites of serpentinization actively occurring today[42]. Simulated temperature is set to 25 °C to model low-temperature continental and subseafloor aquifers where serpentinization is thought to be actively occurring. Compositions of reacting rocks were taken from a precompiled list of Fe-bearing igneous rocks (komatiite, picrite, peridotite, harzburgite, dunite) in the GEOROC database (http://georoc.mpch-mainz.gwdg.de/georoc/). While we did not include the GEOROC precompiled file for lherzolite, the precompiled file for peridotite includes several lherzolites to account for those present in uplifted orogenic massifs[78]. Other rocks in the ultramafic olivine-orthopyroxene-clinopyroxene ternary (e.g., wehrlite, websterite, pyroxenite) were not included as these lithologies comprise a minor component of uplifted ultramafic bodies[78]. Only samples with complete major element data in the GEOROC database were used. Duplicates and samples containing high volatile contents (>1% $CO_2$, S) were removed. Only major element data for MgO, $SiO_2$, $FeO_T$, CaO, $Na_2O$, and $K_2O$, normalized to 100% on an anhydrous basis, were used in the calculations. $FeO_T$ was calculated if it was not reported and if both FeO and $Fe_2O_3$ were indicated ($FeO_T$ = FeO + 0.9$Fe_2O_3$). Reacting rocks include trace amounts of Cl (40 ppm) to yield aqueous solutions with dissolved Cl concentrations ranging from 1 to 10 mmolal, which are similar to those measured from hyperalkaline spring fluids[42]. Overall, a total of 9,414 different Fe-bearing igneous rocks of variable compositions were used for the rock alteration models. Simulations include at least 500 rocks with MgO <10 and >45 wt% and each 5 wt%

interval from 10 to 45 wt%. Further statistical analyses were conducted using these compositional groups. Compositions of all rocks used in the simulations as well as calculated $x_{Fe}$ at different water-to-rock ratios are compiled in Source Data S1.

Thermodynamic data used in the simulations are calculated with the SUPCRT code[79] using standard state thermodynamic data for aqueous species[80,81], together with the revised Helgeson-Kirkham-Flowers equations of state[81]. Data for minerals were mostly taken from Helgeson et al.[82] with the addition of estimated thermodynamic data, consistent with the above database, for Fe(II)-serpentine (greenalite) and talc (minnesotaite) from Wolery and Jove-Colon[83], Fe(III)-serpentine (cronstedtite and hisingerite) from Leong and Shock[41], Fe(II)-brucite from McCollom and Bach[23], and clay minerals from Catalano[84].

**Notes on the rate of rock alteration ($r_{Fe}$).** The rate of serpentinization, $r_{Fe}$, can be informed by experimental work. Recent experiments by McCollom and Donaldsson[77] show that $H_2$-generation rates during low-temperature serpentinization are much slower than rates determined by earlier studies to the point that $H_2$ can be undetectable on the timescales of laboratory experiments. Reaction extents can be quantified through high-temperature experiments where reaction rates are considerably faster. Extrapolation of high-temperature laboratory experiments simulating olivine serpentinization to low temperatures (25 °C) reveals that it takes between $10^6$ and $10^8$ Myrs to completely serpentinize a km$^3$ of ultramafic rock[55], assuming reactive surface areas (e.g., $6 \times 10^6$–$6 \times 10^3$ km$^2$ (km$^3$ rock)$^{-1}$) common in low-temperature continental systems[55]. Multiplying laboratory-determined rates, usually depicted as mole mineral m$^{-2}$ s$^{-1}$ that can be converted to km$^3$ rock km$^{-2}$ yr$^{-1}$, with the reactive surface area (km$^2$ (km$^3$ rock)$^{-1}$) will yield the rate (yr$^{-1}$) required to totally alter a given volume (e.g., 1 km$^3$) of reacting material. Assuming a simplified cubic box model for a 1 km$^3$ ultramafic body, these rates correspond to a serpentinization advance rate between $10^{-8}$ and $10^{-6}$ km yr$^{-1}$.

Alternatively, alteration rates can be derived from field-based investigations at the watershed scale. Using reported weathering rate data from various field-based studies, Navarre-Sitchler and Brantley[85] derived weathering advance rates in basaltic terranes that range from 0.01 to 0.36 $mm^3$ $mm^{-2}$ yr$^{-1}$ (or $10^{-8}$ to $10^{-6.5}$ km yr$^{-1}$). Denudation rates estimated for rocks comprising the Oman ophiolite, which are mostly ultramafic, are in the order of $10^{-6.5}$ km yr$^{-1}$ (ref. [86]), that is similar to the maximum values measured in basaltic environments. The above rates are more or less consistent with the extrapolated values from high-temperature laboratory experiments[55]. However, estimated denudation rates for komatiitic bodies are unknown and laboratory-based experiments are focused on hydrothermal conditions (≥300 °C)[87,88]. In addition, it is uncertain if deep subsurface processes characteristic of serpentinization occur at rates similar to those derived through surficial or shallow sub-surficial weathering processes. Nevertheless, to account for these uncertainties, rates ranging from $10^{-8}$ to $10^{-6}$ km yr$^{-1}$ for serpentinization are used in our models, representing the range of rates observed in both laboratory and field studies.

**Notes on the extent of Fe-bearing igneous rocks ($a_{Fe}$).** The extent to which komatiites and other ultramafic rocks are present in continents, $a_{Fe}$, throughout Earth's history is informed by two types of data: (1) the mass of exposed continents, and (2) how much of the continents are ultramafic in composition. Estimates of the extent of continental exposures during the Archean are highly variable ranging from an extent close to modern levels[63] to a much-reduced presence (e.g., <50% of present level[64,66]). Taylor and McLennan[65] predict lower exposure of continents throughout most of the early Archean followed by a rapid growth at ~3.2 Ga ago until finally attaining around ~70% of present levels by the end of the Archean. Recent reports[56] argue for more gradual growth throughout the Archean, attaining ~60–70% of present levels ~2.5 Ga ago. Our simulations account for all these variations in the estimated mass of continents exposed during the past 3.5 Ga. In addition, recent estimates of the compositions of the continents during the past 3.5 Ga reveal that the crust evolved from being dominantly mafic in composition to one that is predominantly composed of felsic rocks sometime during the Archean[21,52,53]. The extents of komatiitic or ultramafic presence in continents during the Archean were estimated by some of these works[52,53] and are used to calculate the global outgassing rates for $H_2$ in this work. Although these estimates of past ultramafic distribution are based on assumptions using a single bulk rock composition[52,53], natural ultramafic rocks are compositionally variable and can consequently yield varying potentials to generate $H_2$. Our calculations account for these variabilities by including rocks of variable composition ranging from those enriched in Mg to those that are Mg-depleted, using bulk data from the GEOROC database. In addition to the quantities produced in continents, significant amounts of $H_2$ can be sourced from subseafloor serpentinization. However, the distribution of ultramafic rocks in the oceanic crust during the past 3.5 Ga remains unknown. Preliminary discussions on the consequences of various extents of seafloor serpentinization are presented in this work.

Our calculations of the global $H_2$ flux during the past 3.5 Ga are focused on values derived from the hydrous alteration of Fe-bearing rocks and do not account for $H_2$ generation from other sources. In the modern Earth, much of the $H_2$ is sourced from water-rock interactions, particularly from the serpentinization of ultramafic rocks[24]. Aside from water-rock interactions, other sources for abiotic $H_2$ include degassing of magmas, radiolysis of water, and comminution of rocks[24] which are all difficult to constrain for the past 3.5 Ga.

## Data availability

Source data are provided with this paper. Specifically, all data underlying all figures are available in the Source Data tables. Source data are provided with this paper.

## Code availability

Code used in this work (EQ3/6) is freely available from the Lawrence Livermore National Laboratory webpage (https://www-gs.llnl.gov/energy-homeland-security/geochemistry).

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

## Acknowledgements

We would like to thank the ASU Frontiers in Earth System Dynamics (FESD) team for the numerous weekly discussions concerning the Great Oxidation Event. Many years of discussions on rock alteration simulations with various members of the GEOPIG research group have also been helpful. Support is acknowledged from ASU School of Earth and Space Exploration, ASU School of Molecular Sciences, NSF FESD project on Earth System Oxygenation (NSF EAR 1338810, PI: A. Anbar, ASU), NSF Integrated Earth Systems project on the Oman ophiolite (NSF EAR 1515513, PI: E. Shock, ASU; P. Kelemen, Columbia), NASA NExSS Exoplanetary Ecosystems project (NNX15AD53G, PI: S. Desch, ASU) as well as the NASA NAI Rock-Powered Life project (NNA15BB02A, PI: A. Templeton, CU-Boulder).

## Author contributions

J.L. and E.S. designed the project. J.L. and T.E. performed thermodynamic calculations and statistical analyses. J.L. wrote the paper with contributions from all authors. All authors contributed to the discussions of calculation results.

## Competing interests

The authors declare no competing interests.
