## [Peer Review File · Nature Communications]

Decreasing extents of Archean serpentinization contributed to the rise of an oxidized atmosphereREVIEWER COMMENTS

Reviewer #1 (Remarks to the Author):

The authors used equilibrium thermodynamics to model the consumption of O₂ and production of H₂ associated with serpentinization of continental exposures of komatiite and how this would have influenced the Great Oxidation Event in the Paleoproterozoic Era. They suggest that widespread exposures and low-T alteration of Mg-rich komatiite during the Archean limited the buildup of O₂ in the atmosphere. The decreasing formation of komatiite towards the end of the Archean would have allowed O₂ to accumulate in the atmosphere.

Overall, I agree with the conclusions presented in this study: The production of H₂ from serpentinization on early Earth was probably much higher than it is today as the exposure of ultramafic rock (komatiite) was more widespread. Consequently, the production of H₂ decreased with time as less komatiite extruded toward the end of the Archean due to the cooling of the mantle. But it is not clear whether equilibrium thermodynamics can be used to accurately model low-temperature aqueous rock alteration. Did komatiites that experienced low-temperature serpentinization achieve equilibrium? For example, the authors could consider comparing the predicted secondary mineralogy of a komatiite with one that evidently and exclusively underwent low-temperature alteration (without preceding high-temperature alteration). Maybe there are examples in the literature? This could expand on the very general comparison in the supplement.

Another issue is that komatiite exposures in marine settings are not adequately discussed. How would have the much more widespread exposure of komatiite in marine settings affected the timing of the GOE?

If the authors could show that equilibrium can be achieved, at least locally, and that the alteration of komatiite in marine settings would not change their conclusions, that would make this study convincing and likely appropriate for publication in Nature Comms since the idea of serpentinization as a major process in affecting the GOE is an exciting one.

Abstract: Need to say what you actually did - theoretical modeling which is mentioned at the end of the introduction but it's important to mention this earlier.

55: "ferrous iron in rocks into ferric iron in minerals" sounds awkward as both ferrous and ferric iron are present in minerals that make up rocks. change to 'ferrous iron in primary minerals into ferric..."

72-76: Why are you not accounting for magma-poor passive margins and subduction zones where serpentinization can be quite extensive?

79: Not all komatiites are necessarily ultramafic and some can be quite SiO₂-rich. Le Bas (2000, JPet) classification suggests that komatiite can have up to 52 wt.% SiO₂.

89-90: the formation of continents during the Archean is poorly constrained, in particular, the timing of continent formation is debated. Are marine komatiites not considered in this study? If not, please explain why.

101-102: Both of these systems are influenced by serpentinization, however, Rainbow, in particular, is a hybrid mafic-ultramafic hydrothermal system. Some even believe that fluids at Lost City are strongly influenced by interactions with mafic rocks (see e.g., Seyfried et al. 2015, <http://dx.doi.org/10.1016/j.gca.2015.04.040>)

116: The relevance of the application of equilibrium thermodynamics at such low temperatures needs to be explained. Are metastable equilibria considered? It is hard to believe that equilibrium can be attained at such low temperatures without providing specific examples

201: Here, the question is how large or small Archean continents were. 20% would be very significant if the area was as big as today.

214: explain how you converted the reported rate of serpentinization for olivine from $\text{mol m}^{-2} \text{s}^{-1}$ to km yr^{-1} . Note that these rates refer to serpentinization of olivine, not komatiite which is not only composed of olivine. How would this relate to komatiite?

Reviewer #2 (Remarks to the Author):

Leong et al. (2021) details extensive calculations on the H_2 production and thus O_2 drawdown produced by the alteration of continental ultramafic rocks. These calculations are of great interest for constraining the factors influencing the oxygenation of the Archean atmosphere and the timing of the GOE. Overall, the paper is well-argued and the conclusions are clear and relevant. I have only a few questions and concerns that need to be addressed before publication.

Firstly, there is much talk in the paper about komatiites being enriched in the Archean continental crust, but the authors also state that rocks with >35 wt.% MgO are necessary to have a strong reducing effect. There appears to be a disconnect, as the authors argue that Archean olivine cumulates with $>35\%$ MgO were significant in the crust. Most Archean komatiites do not have cumulates that reach this level of MgO content, to my knowledge only South Africa occurrences >3 Ga in age do. Cumulates are also a relatively minor part of many komatiite flows ($<50\%$, often $\sim 30\%$) and are notably absent in cases where pillow structures are evident. I feel that the authors leaned too heavily on Archean komatiites without considering potential peridotite massifs and other ultramafic bodies within the Archean crust. Perhaps simply referring to the all important $>35\%$ MgO bodies as "ultramafic rocks" as opposed to "komatiites" would resolve this problem.

Secondly, it is unclear to me how cumulates, by definition present at the bottom of flows and plutons would come to be exposed at the surface. There is currently no agreement on the existence of plate tectonics in the >3 Ga earth and whether such cumulate bodies would be uplifted and exposed to weathering is unknown. Indeed, many vertical tectonic models of the early earth suppose that cumulate bodies were never uplifted, but simply buried and lost via lower crustal delamination. This will not be a problem if weathering in the Archean crust is expected to reach deep within the crust. The paper needs discussion of how deep within the crust low-temperature serpentinization is expected to occur and whether the weathering front is likely to reach olivine cumulates. An estimate of whether weathering depth has changed between the Archean and the present day would be helpful as well.

Finally, my most fundamental issue with the discussion concerns the discounting of seafloor serpentinization and its effect on Archean hydrogen production and the rise of oxygen. As mantle potential temperatures were significantly higher in the Archean, if plate tectonics existed seafloor spreading is expected to be faster and oceanic crust expected to be more MgO-rich. If vertical tectonics existed instead, seafloor alteration can perhaps be discounted. How significant is H_2 production in ultramafic Archean continents compared to the present-day seafloor alteration flux? If the seafloor flux was removed or increased how would that affect the conclusions of the paper? I am not asking for a full model here as I suspect the authors are saving that work for a future paper, but more discussion and acknowledgement of this issue is much needed in the discussion section of the paper.

Once these problems are addressed, I will be happy to recommend the paper for publication.

-Willie Nicklas
Scripps Institution of Oceanography
University of California, San Diego

Specific In-Line Comments

Line 49: "chemical species as volatiles" sounds awkward. Reword maybe?

Lines 112-113: I find it shocking that no one has even estimated this before. I am glad this study exists as secular reduction in serpentinization is an important piece of the GOE story. Are you sure there were no estimates of these fluxes in any prior papers?

Line 121: A komatiite is defined as a spinifex textured rock with >18% MgO not >20%

Line 122: Picrite is also defined mineralogically by the abundance of olivine phenocrysts and not geochemically. I would suggest changing the names of your categories in the manuscript to high, medium and low MgO lavas as your current categories are technically incorrect.

Line 183: You don't need the second "the"

Line 217: Mention Tang et al. (2019) as an alternate crustal evolution model here, as I wondered if using their model instead would make a difference and had to read down to methods to find out.

Line 237: I have a hard time believing that it is a monotonic decrease, the bulk of komatiite preserved in the rock record and greenstone belt lavas in general show a peak at 2.7 Ga. Would such a large flux of new volcanism lead to O₂ drawdown? It is fairly evident in the greenstone belt record that there were either peaks in komatiite formation (2.7 Ga being the most notable) or periods of enhanced preservation. I do agree that it decreased by the end of the Archean so this does not invalidate your model, but it deserves discussion in the paper.

Line 255: Why are we only considering the continents? Does the model assume that spreading center serpentinization was constant from the Archean to present day? If MOR existed in the Archean higher mantle potential temperatures would potentially lead to significantly faster spreading rates.

Line 365: Remind the reader where the O₂ production rate in the Archean is derived from? Is it from the carbonate C-isotope record and assumptions about the extent of organic C burial?

Lines 380-382: Commentary on seafloor serpentinization processes as they likely operated in the Archean and present day are necessary. I'm not asking for a full model here, just acknowledgement that differences in seafloor spreading rates and MOR temperatures could've lead to dramatically different seafloor alteration hydrogen fluxes in the Archean.

Line 445-446: 533 igneous rocks are unlikely to define the full range of igneous products at any point in Earth's history. Why was the GEOROC database with its tens of thousands of major element analyses not used? We are not talking about trace elements here, where analyses differ in quality, but major element analyses which are likely to be reliable. Why were these specific rocks chosen? Are they especially selected to be representative? Right now the choice seems rather arbitrary. Offer some explanation for the choice or use a broader database please.

Lines 513-515: Need to mention this discussion of seafloor serpentinization in the main manuscript, it is a disservice to leave this important discussion to the methods.

Figure 2: Perhaps comment on why uncertainties become so much larger for higher MgO rocks? Why is this?

Supplement Comments:

Lines 24-25: The "potential" and "amount" singular, not plural

Line 33: Not "yet" fully explained. It will likely be explained at some point.

Figure S2: Again, large uncertainties for select model runs should be explained. It is unclear why they are so much bigger for some calculations.

Response to the Reviewers

We like to thank the reviewers for their comments and helpful suggestions. We have considered and incorporated all of the reviewers' comments and suggestions into the revised manuscript. For reference, we point to associated revisions in the revised main document through line numbers (e.g., Line 100-120). In some cases, we refer to changes introduced to the Supplementary Material by referring to line numbers marked by the letter "S" (e.g., Lines S100-S120). Reviewers' comments are listed first, and our responses follow in italics.

Response to Reviewer 1

The authors used equilibrium thermodynamics to model the consumption of O₂ and production of H₂ associated with serpentinization of continental exposures of komatiite and how this would have influenced the Great Oxidation Event in the Paleoproterozoic Era. They suggest that widespread exposures and low-T alteration of Mg-rich komatiite during the Archean limited the buildup of O₂ in the atmosphere. The decreasing formation of komatiite towards the end of the Archean would have allowed O₂ to accumulate in the atmosphere.

Overall, I agree with the conclusions presented in this study: The production of H₂ from serpentinization on early Earth was probably much higher than it is today as the exposure of ultramafic rock (komatiite) was more widespread. Consequently, the production of H₂ decreased with time as less komatiite extruded toward the end of the Archean due to the cooling of the mantle. But it is not clear whether equilibrium thermodynamics can be used to accurately model low-temperature aqueous rock alteration. Did komatiites that experienced low-temperature serpentinization achieve equilibrium? For example, the authors could consider comparing the predicted secondary mineralogy of a komatiite with one that evidently and exclusively underwent low-temperature alteration (without preceding high-temperature alteration). Maybe there are examples in the literature? This could expand on the very general comparison in the supplement.

Overall, results of simulations showing that serpentine and chlorite as the predominant secondary minerals precipitated during alteration of rocks with high MgO contents are consistent with those observed from extant altered komatiites¹. However, both serpentine and chlorite are stable at a wide range of temperature and pressure conditions^{2,3} and observed mineral paragenesis could record several stages of alteration processes. Some isotopic studies^{4,5} show that some present-day serpentinites in ophiolites can form through interactions with meteoric-derived groundwater under ambient conditions, consistent with equilibrium predictions. Isotopic evidence for komatiites altered at low temperatures are less known. In addition, lower temperature interactions may overprint previous higher temperature products⁶. Thus, evidence for hydrous alteration solely occurring at low temperatures in the komatiite rock record is not well known. Alternatively, attainment of equilibrium even at low-temperature conditions can be supported by studies on the compositions of low-temperature (<40 °C), reduced, and hyperalkaline fluids seeping from ultramafic outcrops in ophiolites, which are present-day analogs of ultramafic-hosted

groundwater during the Archean. Our recent work⁷ shows that while most fluids sampled from hyperalkaline springs in the Oman ophiolite have been modified by shallow subsurficial or surficial processes, some of the most end-member-like fluid (highest pH, lowest dissolved Mg and Si concentrations) are consistent with equilibrium expectations. In that work, which is primarily authored by the lead author of the submitted manuscript, we sampled and analyzed >100 fluids from Oman and demonstrated that equilibrium can be attained even at ambient temperature conditions. The formation of reduced and hyperalkaline fluids during serpentinization can involve several thousands of years⁸, which can ensure the approach to equilibrium even at low-temperature conditions. The above discussions are now added to the supplementary document (**Lines S66-S85**) and briefly in the main manuscript as well (**Lines 116-121**).

Another issue is that komatiite exposures in marine settings are not adequately discussed. How would have the much more widespread exposure of komatiite in marine settings affected the timing of the GOE?

*In the revised manuscript, we added a short section where contributions from seafloor serpentinization are discussed. While recent works^{9,10} have quantified the extents of ultramafic presence in continents, the composition of the oceanic crust during the Archean is less constrained. Nevertheless, we briefly discuss various seafloor settings where H₂ production via oceanic serpentinization can occur and how it can contribute to global H₂ outgassing and O₂ consumption during the Archean. While less constrained compared to its continental counterparts, seafloor serpentinization can supply as much or more H₂ throughout most of the Archean. Likely settings for seafloor serpentinization can occur in oceanic plateaus made by plume-generated large igneous provinces composed of Mg-rich rocks such as komatiites. The decline of komatiites in the seafloor, as well as those emplaced in continents, towards the end of the Archean may have contributed to the accumulation of O₂ in the atmosphere, which is in line with our original conclusions. All the above are discussed in a new section encompassing **Lines 349-394**.*

If the authors could show that equilibrium can be achieved, at least locally, and that the alteration of komatiite in marine settings would not change their conclusions, that would make this study convincing and likely appropriate for publication in Nature Comms since the idea of serpentinization as a major process in affecting the GOE is an exciting one.

Please see above response where we discuss evidence showing equilibrium may be attained even in low-temperature conditions in serpentinizing systems. This paper builds on our extensive analysis of active continental serpentinization in Oman⁷.

Abstract: Need to say what you actually did - theoretical modeling which is mentioned at the end of the introduction but it's important to mention this earlier.

*Abstract is now revised. We mention theoretical modeling as we discussed methods and results in the abstract. See **Line 30**.*

55: “ferrous iron in rocks into ferric iron in minerals” sounds awkward as both ferrous and ferric iron are present in minerals that make up rocks. change to ‘ferrous iron in primary minerals into ferric...’

*Sentence is now revised following the reviewer’s suggestion. See **Lines 55-56**.*

72-76: Why are you not accounting for magma-poor passive margins and subduction zones where serpentinization can be quite extensive?

*Passive margins and subduction zones are now mentioned along with ocean ridge settings and ultramafic bodies in continents (e.g., ophiolites). See **Lines 73-76**.*

79: Not all komatiites are necessarily ultramafic and some can be quite SiO₂-rich. Le Bas (2000, JPet) classification suggests that komatiite can have up to 52 wt.% SiO₂.

*The sentence was meant to be comparative where komatiites are mostly Mg-rich, and Si- and Al-poor relative to basalts and picrites. Nevertheless, we recognize the reviewer’s concern. In the revised manuscript, we add the word “usually” in **Line 134** to denote that not all komatiites are ultramafic in compositions (i.e., Mg-rich, and Si- and Al-poor). We also remove references to specific ranges in MgO content for the rock types as suggested by the second reviewer (see below).*

89-90: the formation of continents during the Archean is poorly constrained, in particular, the timing of continent formation is debated. Are marine komatiites not considered in this study? If not, please explain why.

*Subduction driven by plate tectonics has mostly consumed the Archean seafloor. Please see above response concerning serpentinization of marine komatiites and other rocks. We added a new section in the revised manuscript that briefly discusses seafloor serpentinization as a response to comments by both reviewers. Discussions on relationships between different continental growth models and calculated global H₂ generation and O₂ consumption can be found in the section “Effect of variable extent of ultramafic exposure in continents” following **Line 290**.*

101-102: Both of these systems are influenced by serpentinization, however, Rainbow, in particular, is a hybrid mafic-ultramafic hydrothermal system. Some even believe that fluids at Lost City are strongly influenced by interactions with mafic rocks (see e.g., Seyfried et al. 2015, <http://dx.doi.org/10.1016/j.gca.2015.04.040>)

*We added a phrase that mentions that there might be some mafic contributions to the chemistry of hydrothermal fluids in both Rainbow and Lost City, as proposed by previous work (e.g., Seyfried et al.¹¹) See **Lines 103-104**.*

116: The relevance of the application of equilibrium thermodynamics at such low temperatures needs to be explained. Are metastable equilibria considered? It is hard to believe that equilibrium can be attained at such low temperatures without providing specific examples

*See response above. We supply evidence that supports the attainment of equilibrium during low-temperature serpentinization. See **Lines 116-121** in the main manuscript and **Lines S66-S85** in the supplementary document.*

201: Here, the question is how large or small Archean continents were. 20% would be very significant if the area was as big as today.

*We dedicate a section that expounds on how different continental growth models affect calculated O₂-consumption rates. See sections entitled “Effect of variable extent of komatiite exposure in continents” starting from **Line 290**. Considerations of different continental growth models (e.g., ref. 12–14) are also shown in Figure 4. Overall, model results show H₂ generation via continental serpentinization can be significant if continents are at least 60% of present levels during the mid-Archean (3.0 Ga), consistent with the recent models of Dhuime et al.¹² H₂ generation through serpentinization is insignificant during this time period if scenarios depicting less continental presence proposed by earlier works^{14–16} are considered. See **Lines 299-314**.*

214: explain how you converted the reported rate of serpentinization for olivine from mol m⁻² s⁻¹ to km yr⁻¹. Note that these rates refer to serpentinization of olivine, not komatiite which is not only composed of olivine. How would this relate to komatiite?

*In the methods section (**Lines 493-507**), we show how we arrive to serpentinization rates with units of km yr⁻¹ from laboratory-derived rates (usually quantified as mol mineral m⁻² s⁻¹ that can be converted into km³ mineral or rock km⁻² yr⁻¹) following assumptions about the reactive surface area. While there are laboratory experiments simulating komatiite serpentinization at hydrothermal conditions, there is yet to be a study focused on low-temperature komatiite alteration. There are experiments for olivine or peridotite serpentinization, closest analogs for komatiites, at low-temperature conditions but results of these experiments have been questioned¹⁷. Extrapolation of high-temperature experiments simulating olivine serpentinization¹⁸ to low-temperatures, by consideration of reactive surface areas, are consistent with rates derived from watershed-scale studies on ultramafic and basaltic environments. We used these rates (10⁻⁸ to 10⁻⁶ km yr⁻¹) in our calculations. Only the maximum rate yields substantial H₂ generation. While this represents an upper limit on rates perceived to occur in modern low-temperature ultramafic aquifers¹⁸, such a rate could have been normal under warmer and more dynamic Archean surface conditions.*

Response to Reviewer 2

Leong et al. (2021) details extensive calculations on the H₂ production and thus O₂ drawdown produced by the alteration of continental ultramafic rocks. These calculations are of great interest for constraining the factors influencing the oxygenation of the Archean atmosphere and the timing of the GOE. Overall, the paper is well-argued and the conclusions are clear and relevant. I have only a few questions and concerns that need to be addressed before publication.

Firstly, there is much talk in the paper about komatiites being enriched in the Archean continental crust, but the authors also state that rocks with >35 wt.% MgO are necessary to have a strong reducing effect. There appears to be a disconnect, as the authors argue that Archean olivine cumulates with >35% MgO were significant in the crust. Most Archean komatiites do not have cumulates that reach this level of MgO content, to my knowledge only South Africa occurrences >3 Ga in age do. Cumulates are also a relatively minor part of many komatiite flows (<50%, often ~30%) and are notably absent in cases where pillow structures are evident. I feel that the authors leaned too heavily on Archean komatiites without considering potential peridotite massifs and other ultramafic bodies within the Archean crust. Perhaps simply referring to the all important >35% MgO bodies as “ultramafic rocks” as opposed to “komatiites” would resolve this problem.

The reviewer has a good point. We have rewritten most references to komatiites, particularly when referring to those with high MgO contents, into ultramafic. The title now refers to “Archean serpentization”, instead of “Archean komatiite serpentization”. The title is now more inclusive of the rock type and composition (i.e., any rock that can serpentize including komatiites and other ultramafic rocks). Overall, in the revised manuscript, most of the occurrences of the word komatiite/komatiitic was replaced (from ~90 to ~30 words) with ultramafic. In addition, we significantly expanded the number of simulated rock compositions from 533 to 9,414 to better capture the full range of igneous products at any point of Earth’s history.

Secondly, it is unclear to me how cumulates, by definition present at the bottom of flows and plutons would come to be exposed at the surface. There is currently no agreement on the existence of plate tectonics in the >3 Ga earth and whether such cumulate bodies would be uplifted and exposed to weathering is unknown. Indeed, many vertical tectonic models of the early earth suppose that cumulate bodies were never uplifted, but simply buried and lost via lower crustal delamination. This will not be a problem if weathering in the Archean crust is expected to reach deep within the crust. The paper needs discussion of how deep within the crust low-temperature serpentization is expected to occur and whether the weathering front is likely to reach olivine cumulates. An estimate of whether weathering depth has changed between the Archean and the present day would be helpful as well.

*Cumulates are indeed often associated with deeper portions of komatiite flows. H₂-rich seeps are likely to be sourced from deep-seated aquifers in komatiitic bodies where fluid interactions with cumulate rocks are more likely. Fluids can indeed infiltrate deep into the Earth's crust. Geophysical surveys¹⁹ reveal that serpentinized peridotites can occur several kilometers below the seafloor. Other deep-seated settings where seawater-derived fluids can interact with ultramafic rocks, via deep-seated geological structures, are in passive margins and subduction zones²⁰. Reduced fluids were also observed in deep-seated mines in continental cratons²¹. However, the extent to which these processes occurred during the Archean is unknown. We incorporate some of the above statements into the revised manuscript. See **Lines 316-318**.*

*Alternatively, komatiite bodies are thought to be formed in plume-generated settings in the seafloor such as oceanic plateaus²². Deep-seated cumulate rocks may readily react with groundwater during and after continental emplacement like in modern ophiolitic bodies where peridotites, which are likewise located deep in the oceanic lithosphere sequence, are now exposed to surface conditions via tectonic processes and enable the generation of most of present-day H₂-rich, hyperalkaline springs. Exposures of mantle rocks often dominate ophiolitic bodies, such as in the Oman ophiolite where peridotites comprise >50% of exposed lithologies²³. We incorporate some of the above statements in the revised manuscript. See **Lines 318-321**.*

*As the reviewer mentioned, komatiite flows are likely to be compositionally heterogenous, where cumulate rocks comprise <50% of most flows. We conducted further simulations where H₂-generation potential is calculated for a heterogenous igneous province comprised of variable mixtures of mafic (MgO <10%) and relatively more Mg-rich lithologies. Results of ~1,000,000 mixing calculations are shown as colored symbols and curves in the new Figure S2a and S2b, respectively. As H₂ production in rocks with MgO >35% is far more substantial than in rocks with lower Mg contents, at least 40% (dark-blue to violet curves in Figure S2b) of a given quantity of fluids present in mafic-ultramafic igneous provinces encompassing 18 km² of seafloor (orange line in Figure S2, depicting range of oceanic plateaus presence in the present day) would need to interact with rocks with MgO >35% to still result in consumption of >2 Tmole O₂ yr⁻¹. A hotter early Earth would likely result in more extensive occurrences of plume-generated settings than present (i.e., >18 km²), which would increase the likelihood of offsetting production of >2 Tmole O₂ yr⁻¹. The above statements are incorporated in the manuscript. See **Lines 384-394**. Additional details on the mixing calculations can be found in the supplementary document, see **Lines S138-S160**.*

Finally, my most fundamental issue with the discussion concerns the discounting of seafloor serpentinization and its effect on Archean hydrogen production and the rise of oxygen. As mantle potential temperatures were significantly higher in the Archean, if plate tectonics existed seafloor spreading is expected to be faster and oceanic crust expected to be more MgO-rich. If vertical tectonics existed instead, seafloor alteration can perhaps be discounted. How significant is H₂

production in ultramafic Archean continents compared to the present-day seafloor alteration flux? If the seafloor flux was removed or increased how would that affect the conclusions of the paper? I am not asking for a full model here as I suspect the authors are saving that work for a future paper, but more discussion and acknowledgement of this issue is much needed in the discussion section of the paper.

Once these problems are addressed, I will be happy to recommend the paper for publication.

A new section that briefly discusses seafloor serpentinization is now included in the revised manuscript. In this section we discuss various marine settings (ridges, oceanic plateaus) that can likely host ultramafic rocks. We edited Figure 4 to make it inclusive of both continental and seafloor serpentinization. The extent of ultramafic presence in both continents and seafloor that is required to yield significant H₂ outgassing and O₂ consumption (>2 and >3 Tmole O₂ yr⁻¹) is depicted in the revised Figure 4. The nature of ridges formed in divergent tectonic setting during the Archean remains unknown. In modern settings, seafloor serpentinization is usually associated with slow-spreading environments where mantle-derived ultramafic rocks are exposed through detachment faults²⁴. Faster spreading²⁵ and thicker oceanic crusts²⁶ might not promote emplacement of deep-seated Mg-rich ultramafic rocks into the surface where they can readily react with seawater. Alternatively, H₂ generation in the Archean seafloor might be associated with plume-generated large igneous provinces forming oceanic plateaus. In the revised manuscript, we show that if the extent of oceanic plateaus during the Archean is similar to that of the present (18 million km²), H₂ generation can be significant provided that the oceanic plateaus are composed of Mg-rich ultramafic rocks.

*The new section where we discuss oceanic serpentinization is found in **Lines 349-394** in the revised manuscript.*

Specific In-Line Comments

Line 49: “chemical species as volatiles” sounds awkward. Reword maybe?

*The sentence is now reworded. Please see **Lines 48-50**.*

Lines 112-113: I find it shocking that no one has even estimated this before. I am glad this study exists as secular reduction in serpentinization is an important piece of the GOE story. Are you sure there were no estimates of these fluxes in any prior papers?

Existing global estimates of serpentinization-derived H₂ pertain to modern settings^{21,24,27,28}. Past serpentinization-derived H₂ has been modelled but only back to the last 200 Myrs²⁷. Relationships between rock compositions and H₂ generation are also less known prior to this work. While there are works^{29,30} that estimate H₂ generation potentials of Fe-bearing igneous rocks, modelled

rock compositions are limited ($N < 50$) and are mostly focused on peridotites and basalts. Our original manuscript modelled 533 rock compositions, which is significantly more than existing studies. In the revised manuscript, we modelled 9,414 rocks with compositions ranging from ultramafic to mafic.

We revised the text to emphasize that simulations of ultramafic and mafic rock compositions exist and this work improves on previous models through simulation of a significantly more comprehensive pool of rock compositions (**Lines 111-116**).

Line 121: A komatiite is defined as a spinifex textured rock with $>18\%$ MgO not $>20\%$

The sentence is now revised following the above comment. See **Line 236**.

Line 122: Picrite is also defined mineralogically by the abundance of olivine phenocrysts and not geochemically. I would suggest changing the names of your categories in the manuscript to high, medium and low MgO lavas as your current categories are technically incorrect.

In the revised manuscript, we removed all compositional range that depicts each rock type, following the reviewer's suggestions (See **Lines 125-129**). In depicting H_2 -generation and O_2 -consumption potentials, we now specifically indicate the compositional group (e.g., >35 wt% MgO, 30–35 wt% MgO) instead of specific rock name.

Line 183: You don't need the second "the"

Deleted. See **Line 190**.

Line 217: Mention Tang et al. (2019) as an alternate crustal evolution model here, as I wondered if using their model instead would make a difference and had to read down to methods to find out.

That sentence pertaining to Tang et al.⁹ in the methods section is now moved into the discussion as recommended by the reviewer. See **Lines 224-228**.

Line 237: I have a hard time believing that it is a monotonic decrease, the bulk of komatiite preserved in the rock record and greenstone belt lavas in general show a peak at 2.7 Ga. Would such a large flux of new volcanism lead to O_2 drawdown? It is fairly evident in the greenstone belt record that there were either peaks in komatiite formation (2.7 Ga being the most notable) or periods of enhanced preservation. I do agree that it decreased by the end of the Archean so this

does not invalidate your model, but it deserves discussion in the paper.

*The reviewer makes a good point. We now include a brief discussion that a decrease in O₂ consumption potentials from 3 to 2.5 Ga might not be monotonic if there is an increase in komatiite volcanism, especially at 2.7 Ga as shown by the rock record. We also note that komatiite occurrence significantly declined by the end of the Archean so model scenarios remain the same. See **Lines 250-255**.*

Line 255: Why are we only considering the continents? Does the model assume that spreading center serpentinization was constant from the Archean to present day? If MOR existed in the Archean higher mantle potential temperatures would potentially lead to significantly faster spreading rates.

See above comments. We added a new section that discusses seafloor serpentinization.

Line 365: Remind the reader where the O₂ production rate in the Archean is derived from? Is it from the carbonate C-isotope record and assumptions about the extent of organic C burial?

*We added a sentence that indicates how predictions of O₂ production rate in the Early Earth are based on the C-isotope record as well as on how extensive organic C burial was³¹. See **Lines 280-283**.*

Lines 380-382: Commentary on seafloor serpentinization processes as they likely operated in the Archean and present day are necessary. I'm not asking for a full model here, just acknowledgment that differences in seafloor spreading rates and MOR temperatures could've lead to dramatically different seafloor alteration hydrogen fluxes in the Archean.

See above comments. We added a new section that discusses seafloor serpentinization.

Line 445-446: 533 igneous rocks are unlikely to define the full range of igneous products at any point in Earth's history. Why was the GEOROC database with its tens of thousands of major element analyses not used? We are not talking about trace elements here, where analyses differ in quality, but major element analyses which are likely to be reliable. Why were these specific

rocks chosen? Are they especially selected to be representative? Right now the choice seems rather arbitrary. Offer some explanation for the choice or use a broader database please.

We retrieved the original 533 simulated rock compositions from several studies focused on igneous processes. In doing so, we thought that we were using the 'freshest' rocks available in the literature. The sample size is based on the computational limits available to the main author during conceptualization and early stages of the project. The sampling pool also results in compositional groups (e.g., wt% MgO ranging from 30 to 35%) with population of at least 30 rocks that is adequate for statistical analyses. This sample size ($N = 533$) is significantly higher than existing thermodynamic models (usually $N < 50$, e.g., refs. ^{29,30}). However, the reviewer has a point that this set of samples might not define the full range of igneous rocks with compositions from ultramafic to mafic. In the revised manuscript, we significantly expanded the number of modelled rock compositions from 533 to 9,414. All the new modelled rocks were taken from precompiled files stored in the GEOROC database, as recommended by the reviewer. The new calculations are thus more extensive, and, more importantly, more inclusive in terms of modelled rock compositions. Statistical analyses of results of simulations are also more robust with the expansion of modelled rock compositions (N for every compositional group is at least 500). There will be some minor changes in the values (e.g., calculated H_2 -generation and O_2 -consumption potentials during alteration of each type of rocks) indicated in the text but the overall trends are closely similar and thus the manuscript's narrative and conclusions remain the same. The expansion of our computational work is possible with the help of co-author Dr. Tucker Ely whose work in high-throughput thermodynamic simulations greatly benefited the revised manuscript.

Lines 513-515: Need to mention this discussion of seafloor serpentinization in the main manuscript, it is a disservice to leave this important discussion to the methods.

See above comments. We now add a new section that discusses seafloor serpentinization.

Figure 2: Perhaps comment on why uncertainties become so much larger for higher MgO rocks? Why is this?

*A brief discussion on the reason for the high variations in H_2 -generation potentials for rocks belonging to high MgO compositional groups is added in the revised manuscript. See **Lines 197-204**.*

Supplement Comments:

Lines 24-25: The “potential” and “amount” singular, not plural

Revised. See Lines S36-S37.

Line 33: Not “yet” fully explained. It will likely be explained at some point.

Revised. See Line S45.

Figure S2: Again, large uncertainties for select model runs should be explained. It is unclear why they are so much bigger for some calculations.

The new Figure S2 now includes distribution curves for each compositional group, instead of only displaying the mean and percentiles as in the original Figure S2. The discussions following the revised figure expound on the large variations in the H₂-generation potentials of rocks within certain compositional groups. See Lines S86-S137.

References:

1. Arndt, N. T., Leshner, C. M. & Barnes, S. J. Mineralogy. in *Komatiite* 98–129 (Cambridge University Press, 2008). doi:10.1017/CBO9780511535550.005.
2. Evans, B. W. The serpentinite multisystem revisited: Chrysotile is metastable. *Int. Geol. Rev.* **46**, 479–506 (2004).
3. de Caritat, P., Hutcheon, I. & Walshe, J. L. Chlorite Geothermometry: A Review. *Clays Clay Miner.* **41**, 219–239 (1993).
4. Barnes, I., O’Neil, J. R. & Trescases, J. J. Present day serpentinization in New Caledonia, Oman and Yugoslavia. *Geochim. Cosmochim. Acta* **42**, 144–145 (1978).

5. Sturchio, N. C., Abrajano, T. A., Murowchick, J. B. & Muehlenbachs, K. Serpentinization of the Acoje massif, Zambales ophiolite, Philippines: hydrogen and oxygen isotope geochemistry. *Tectonophysics* **168**, 101–107 (1989).
6. Kyser, T. K., O’Hanley, D. S. & Wicks, F. J. The origin of fluids associated with serpentinization; evidence from stable-isotope compositions. *Can. Mineral.* **37**, 223–237 (1999).
7. Leong, J. A. M. *et al.* Theoretical predictions versus environmental observations on serpentinization fluids: Lessons from the Samail ophiolite in Oman. *J. Geophys. Res. Solid Earth* **126**, e2020JB020756 (2021).
8. Paukert Vankeuren, A. N., Matter, J. M., Stute, M. & Kelemen, P. B. Multitracer determination of apparent groundwater ages in peridotite aquifers within the Samail ophiolite, Sultanate of Oman. *Earth Planet. Sci. Lett.* **516**, 37–48 (2019).
9. Tang, M., Chen, K. & Rudnick, R. L. Archean upper crust transition from mafic to felsic marks the onset of plate tectonics. *Science* **351**, 372–375 (2016).
10. Greber, N. D. *et al.* Titanium isotopic evidence for felsic crust and plate tectonics 3.5 billion years ago. *Science* **357**, 1271–1274 (2017).
11. Seyfried, W. E., Pester, N. J., Tutolo, B. M. & Ding, K. The Lost City hydrothermal system: Constraints imposed by vent fluid chemistry and reaction path models on subseafloor heat and mass transfer processes. *Geochim. Cosmochim. Acta* **163**, 59–79 (2015).
12. Dhuime, B., Hawkesworth, C. J., Cawood, P. A. & Storey, C. D. A change in the geodynamics of continental growth 3 billion Years Ago. *Science* **335**, 1334–1336 (2012).
13. Armstrong, R. L. Radiogenic isotopes: The case for crustal recycling on a near-steady-state no-continental-growth Earth. *Philos. Trans. R. Soc. Lond. Ser. Math. Phys. Sci.* **301**, 443–472 (1981).

14. Taylor, S. R. & McLennan, S. M. *The continental crust: its composition and evolution: an examination of the geochem. record preserved in sedimentary rocks.* (Blackwell, 1985).
15. Condie, K. C. & Aster, R. C. Episodic zircon age spectra of orogenic granitoids: The supercontinent connection and continental growth. *Precambrian Res.* **180**, 227–236 (2010).
16. Allègre, C. J. & Rousseau, D. The growth of the continent through geological time studied by Nd isotope analysis of shales. *Earth Planet. Sci. Lett.* **67**, 19–34 (1984).
17. McCollom, T. M. & Donaldson, C. Generation of hydrogen and methane during experimental low-temperature reaction of ultramafic rocks with water. *Astrobiology* **16**, 389–406 (2016).
18. Lamadrid, H. M., Zajacz, Z., Klein, F. & Bodnar, R. J. Synthetic fluid inclusions XXIII. Effect of temperature and fluid composition on rates of serpentinization of olivine. *Geochim. Cosmochim. Acta* **292**, 285–308 (2021).
19. Carlson, R. L. The abundance of ultramafic rocks in Atlantic Ocean crust. *Geophys. J. Int.* **144**, 37–48 (2001).
20. Klein, F., Tarnas, J. D. & Bach, W. Abiotic sources of molecular hydrogen on Earth. *Elements* **16**, 19–24 (2020).
21. Sherwood Lollar, B., Onstott, T. C., Lacrampe-Couloume, G. & Ballentine, C. J. The contribution of the Precambrian continental lithosphere to global H₂ production. *Nature* **516**, 379–382 (2014).
22. Storey, M., Mahoney, J. J., Kroenke, L. W. & Saunders, A. D. Are oceanic plateaus sites of komatiite formation? *Geology* **19**, 376–379 (1991).
23. Nicolas, A., Boudier, F., Ildefonse, B. & Ball, E. Accretion of Oman and United Arab Emirates ophiolite – Discussion of a new structural map. *Mar. Geophys. Res.* **21**, 147–180 (2000).

24. Cannat, M., Fontaine, F. & Escartín, J. Serpentinization and associated hydrogen and methane fluxes at slow spreading ridges. in *Geophysical Monograph Series* (eds. Rona, P. A., Devey, C. W., Dymant, J. & Murton, B. J.) vol. 188 241–264 (American Geophysical Union, 2010).
25. Bickle, M. J. Heat loss from the earth: A constraint on Archaean tectonics from the relation between geothermal gradients and the rate of plate production. *Earth Planet. Sci. Lett.* **40**, 301–315 (1978).
26. Sleep, N. H. & Windley, B. F. Archean plate tectonics: Constraints and inferences. *J. Geol.* **90**, 363–379 (1982).
27. Merdith, A. S. *et al.* Pulsated global hydrogen and methane flux at mid-ocean ridges driven by Pangea breakup. *Geochem. Geophys. Geosystems* **21**, e2019GC008869 (2020).
28. Sleep, N. H. & Bird, D. K. Niches of the pre-photosynthetic biosphere and geologic preservation of Earth's earliest ecology. *Geobiology* **5**, 101–117 (2007).
29. Wetzel, L. R. & Shock, E. L. Distinguishing ultramafic-from basalt-hosted submarine hydrothermal systems by comparing calculated vent fluid compositions. *J. Geophys. Res. Solid Earth* **105**, 8319–8340 (2000).
30. Klein, F., Bach, W. & McCollom, T. M. Compositional controls on hydrogen generation during serpentinization of ultramafic rocks. *Lithos* **178**, 55–69 (2013).
31. Krissansen-Totton, J., Kipp, M. A. & Catling, D. C. Carbon cycle inverse modeling suggests large changes in fractional organic burial are consistent with the carbon isotope record and may have contributed to the rise of oxygen. *Geobiology* **19**, 342–363 (2021).

REVIEWERS' COMMENTS

Reviewer #1 (Remarks to the Author):

The authors did a fine job revising the manuscript and addressed all my concerns. This is a timely study on an important topic that should be well-received by the readership of Nature Communications.

Please find below a list of minor line-specific comments that I'd like to see addressed before publication.

58-61: Recent work (<https://doi.org/10.1016/j.gca.2021.05.048>) has shown that the alteration of granite can produce significant amounts of H₂. Consider revising this statement accordingly.

72: change to "the modern geological supply" as much of the H₂ in Earth's atmosphere stems from the combustion of fossil fuels.

78: Add a reference to substantiate this claim

89: add reference and link for the GEOROC database

136: dunite makes up only a small fraction of ultramafic rocks in the oceanic mantle

146: Ref 22 is a theoretical modeling study that did not examine the occurrence of NiFe alloys in serpentinite. Swap that reference with this one <https://doi.org/10.1093/petrology/egn071>

179: consider adding ref <https://doi.org/10.1016/j.gca.2014.07.002>

204: consider adding ref <https://doi.org/10.1016/j.gca.2009.08.021> after unoxidized

258-259, also 337: Tutolo et al (ref 41) suggest a diminished potential for H₂ production for serpentinization of komatiite due to elevated concentrations of Si in sea water. How would this affect your calculations?

395: consider changing this to "Cessation of serpentinization as a driver for the GOE" or something alike

463: add or change ref 68 to <https://doi.org/10.1089/ast.2015.1382>

468-470: would these conditions be also relevant to sea floor environments?

471: no lherzolite wehrlite? What about pyroxenite? Were those left out?

Reviewer #2 (Remarks to the Author):

The authors have adequately and thoroughly addressed all my concerns, and have performed new calculations using a much larger set of bulk-rock compositions. I have no more comments and think it should be published.

-Robert W. Nicklas

Response to the Reviewers

We would like to thank the reviewers for their comments and helpful suggestions. We have considered and incorporated all of the reviewers' comments and suggestions into the revised manuscript. For reference, we point to associated revisions in the revised main document through line numbers (e.g., Line 100-120). Reviewers' comments are listed first, and our responses follow in italics.

Responses to Reviewer #1

The authors did a fine job revising the manuscript and addressed all my concerns. This is a timely study on an important topic that should be well-received by the readership of Nature Communications.

Please find below a list of minor line-specific comments that I'd like to see addressed before publication.

We would like to thank the reviewer for the helpful comments and suggestions. The additional references suggested by reviewer (see below) are extremely helpful. We also included additional statements, as pointed out by the reviewer, that improved our manuscript.

58-61: Recent work (<https://doi.org/10.1016/j.gca.2021.05.048>) has shown that the alteration of granite can produce significant amounts of H₂. Consider revising this statement accordingly.

In this section, we discussed recent work by Lee et al. (2016) which showed that the transition in continents enriched in Fe-rich mafic rocks to those dominated by Fe-poor felsic rocks resulted in a decreased in the reductive efficiency of continents via direct consumption of O₂ during oxidation of Fe⁺² in the rocks (Rxn 1). It is not related to H₂ production (Rxn 2) which we discussed in the following paragraph. However, we recognize the recent work pointed out by the reviewer. We added a few sentences alluding to this recent work that shows that the hydrothermal alteration of felsic rocks can also yield high amounts of H₂. However, it is unknown if this extends to low-temperature conditions which is the focus of the manuscript. See Lines 73-78.

72: change to "the modern geological supply" as much of the H₂ in Earth's atmosphere stems from the combustion of fossil fuels.

Thank you for pointing this out. We changed the text as recommended by the reviewer. See Line 73.

78: Add a reference to substantiate this claim

We added a new reference (Condie and O’Niell, 2010) as suggested by the reviewer. See Line 83.

89: add reference and link for the GEOROC database

We added a reference (Sarbas and Nohl, 2008) as well as the link to the GEOROC database. See Lines 95-96.

136: dunite makes up only a small fraction of ultramafic rocks in the oceanic mantle

We removed “dunite” from the statement, as pointed out by the reviewer. See Line 145.

146: Ref 22 is a theoretical modeling study that did not examine the occurrence of NiFe alloys in serpentinite. Swap that reference with this one <https://doi.org/10.1093/petrology/egn071>

Thank you for pointing this out. We replaced the reference with Klein and Bach (2009) as recommended by the reviewer. See Line 156.

179: consider adding ref <https://doi.org/10.1016/j.gca.2014.07.002>

We added Plümper et al. (2014) as recommended by the reviewer. See Line 191.

204: consider adding ref <https://doi.org/10.1016/j.gca.2009.08.021> after unoxidized

We added the recommended article (Klein et al., 2009). See Line 218.

258-259, also 337: Tutolo et al (ref 41) suggest a diminished potential for H₂ production for serpentinization of komatiite due to elevated concentrations of Si in sea water. How would this affect your calculations?

In the revised methods section (Lines 490 – 503), we add brief discussions on the possible influences of reacting fluids with variable starting compositions. In the current work, we only modeled the effects of variable dissolved O₂ concentration of starting fluids. In the previous version of the manuscript, we only discussed why we did not vary the starting dissolved CO₂ (or inorganic carbon) due to lack of data for Archean groundwater and seawater compositions. In the revised manuscript, we added a statement that in addition to dissolved

CO₂, other solutes in reacting fluids such as dissolved Si and sulfate can likely influence the redox state of fluids following Tutolo et al. (2020) and Ely (2020), respectively. However, the relationship between the starting fluid concentrations and resulting alteration minerals and fluids at various extents of reaction (i.e., water-rock ratios) remain unconstrained. It is likely that at the very low water-rock conditions (e.g., W/R = 1), which is the focus of this work, the compositions of the reacting rock exert more influence than the starting composition of the reacting fluid. We have shown this in the case for dissolved O₂ where we modeled the effect of fluids with starting values from low to high (i.e., fluid in equilibrium with an atmosphere with log pO₂ of -5 to -0.7). Results of calculations are depicted in Supplementary Figure S4, which shows that reaction paths converge when W/R < 100. As a consequence, the resulting H₂ potentials are similar for a given rock composition despite reaction with fluids with variable dissolved O₂ content. Similarly, the Si concentration can likewise be buffered to fixed values where serpentine coexists and is in equilibrium with brucite or talc, as shown in Figure 1, which depicts results of alteration models at W/R = 1. Future simulations can disentangle the relationship between H₂ generation, W/R ratios, and the starting composition of reacting groundwater or seawater.

395: consider changing this to “Cessation of serpentinization as a driver for the GOE” or something alike

We adapted the reviewer’s suggestions in the revised manuscript. See Line 417.

463: add or change ref 68 to <https://doi.org/10.1089/ast.2015.1382>

We added McCollom and Donaldsson (2016) in the revised manuscript, as recommended by the reviewer. See Line 489.

468-470: would these conditions be also relevant to sea floor environments?

Thank you for pointing this out. We now added “subseafloor aquifers” into the revised manuscript. See Line 507.

471: no Iherzolite wehrlite? What about pyroxenite? Were those left out?

While we did not include the GEOROC precompiled file for Iherzolite, the precompiled file for peridotite includes several Iherzolites to account for those present in uplifted orogenic massifs. Other rocks in the ultramafic olivine-orthopyroxene-clinopyroxene ternary (e.g., wehrlite, websterite, pyroxenite) were not included as these lithologies comprise a minor component of uplifted ultramafic bodies (Bodinier and Godard, 2003). The above statements have been added into the revised manuscript. See Lines 511 – 516.

Responses to Reviewer #2

The authors have adequately and thoroughly addressed all my concerns, and have performed new calculations using a much larger set of bulk-rock compositions. I have no more comments and think it should be published.

-Robert W. Nicklas

We would like to thank Dr. Nicklas for his very helpful comments and suggestions. The revisions following his suggestion to use a much larger bulk-rock database has greatly improved our manuscript.